# Non-Stationary Online Structured Prediction with Surrogate Losses

**Shinsaku Sakaue** [1 2 3]  **Han Bao** [4 5 3]  **Yuzhou Cao** [6]

## Abstract

Online structured prediction, including online classification as a special case, is the task of sequentially predicting labels from input features. In this setting, the *surrogate regret*—the cumulative excess of the actual target loss (e.g., the 0-1 loss) over the surrogate loss (e.g., the logistic loss) incurred by the best fixed estimator—has gained attention because it admits a finite bound independent of the time horizon $T$. However, such guarantees break down in *non-stationary* environments, where every fixed estimator may incur surrogate loss that grows linearly with $T$. To address this limitation, we obtain an upper bound of $F_T + O(1+P_T)$ on the cumulative target loss, where $F_T$ is the cumulative surrogate loss of any comparator sequence and $P_T$ is its *path length*. This bound depends on $T$ only through $F_T$ and $P_T$, thus offering stronger guarantees under non-stationarity. Our core idea is to combine the dynamic regret analysis of online gradient descent (OGD) with the *exploit-the-surrogate-gap* technique. This viewpoint sheds light on the usefulness of a Polyak-style learning rate for OGD, which systematically yields target-loss bounds and performs well empirically. We then extend our approach to broader settings beyond prior work via the *convolutional Fenchel–Young loss*. Finally, a lower bound shows that the dependence on $F_T$ and $P_T$ is tight.

## 1. Introduction

In supervised learning, the goal is to predict the ground-truth label $y \in \mathcal{Y}$ given an input vector $x \in \mathcal{X}$. For example,

the $K$-class classification problem asks to predict a label from $\mathcal{Y} = \{1, \ldots, K\}$. Beyond multiclass classification, structured prediction also covers problems with combinatorial outputs, including multilabel classification and ranking. Such problems are collectively known as *structured prediction*, which has been extensively studied and applied to various fields, including natural language processing and bioinformatics (BakIr et al., 2007).

We study the sequential version of structured prediction, where a learner and an environment interact sequentially over $T$ rounds. At each round $t$, the learner observes input $x_t \in \mathcal{X}$ and makes prediction $\hat{y}_t \in \hat{\mathcal{Y}}$, where $\hat{\mathcal{Y}}$ is the set of predictions. The environment then reveals ground-truth $y_t \in \mathcal{Y}$, and the learner incurs target loss $\ell(\hat{y}_t, y_t)$, which measures the discrepancy between $\hat{y}_t$ and $y_t$. The learner's goal is to minimize the cumulative target loss over $T$ rounds. A typical example is online classification, where $\mathcal{Y} = \hat{\mathcal{Y}} = \{1, \ldots, K\}$ and the target loss is the 0-1 loss.

In many structured prediction problems, the label set $\mathcal{Y}$ is discrete, which prevents us from learning direct mappings from $\mathcal{X}$ to $\mathcal{Y}$ via efficient continuous optimization methods. A standard workaround is to use the surrogate loss framework (see, e.g., Bartlett et al. 2006). Specifically, we define an intermediate score space $\mathbb{R}^d$ between $\mathcal{X}$ and $\mathcal{Y}$ and a surrogate loss $L(\theta_t, y_t)$,[1] which quantifies how well the score vector $\theta_t \in \mathbb{R}^d$ is aligned with ground-truth $y_t \in \mathcal{Y}$. With this framework, we can efficiently learn mappings from $\mathcal{X}$ to $\mathbb{R}^d$ using continuous optimization. Following prior work (Van der Hoeven, 2020; Van der Hoeven et al., 2021; Sakaue et al., 2024; Shibukawa et al., 2025), we focus on learning linear estimators $W_t \colon \mathcal{X} \to \mathbb{R}^d$ over $t = 1, \ldots, T$, where each $W_t$ computes a score vector as $\theta_t = W_t x_t$. We make a prediction $\hat{y}_t \in \hat{\mathcal{Y}}$ from the score vector $\theta_t \in \mathbb{R}^d$ via a mapping, called *decoding*, from $\mathbb{R}^d$ to $\hat{\mathcal{Y}}$; this depends on the problem setting as detailed in Sections 3.1 and 4.1.

In online structured prediction with the surrogate loss framework, a widely used performance measure is the *surrogate regret*, denoted by $R_T$:

$$\sum_{t=1}^{T} \ell(\hat{y}_t, y_t) = \sum_{t=1}^{T} L(U x_t, y_t) + R_T, \qquad (1)$$

---

[1]CyberAgent, Tokyo, Japan [2]National Institute of Informatics, Tokyo, Japan [3]Center for Advanced Intelligence Project, RIKEN, Tokyo, Japan [4]The Institute of Statistical Mathematics, Tokyo, Japan [5]Tohoku University, Miyagi, Japan [6]Nanyang Technological University, Singapore. Correspondence to: Shinsaku Sakaue <shinsaku.sakaue@gmail.com>, Han Bao <bao.han@ism.ac.jp>, Yuzhou Cao <nanjing.caoyuzhou@gmail.com>.

*Proceedings of the 43rd International Conference on Machine Learning*, Seoul, South Korea. PMLR 306, 2026. Copyright 2026 by the author(s).

---

[1]While we focus on the Euclidean score space, a general framework on Hilbert spaces is also studied (Ciliberto et al., 2016, 2020).

where $\boldsymbol{U}$ is the best offline linear estimator minimizing the cumulative surrogate loss.[2] The surrogate regret $R_T$ represents the learner's extra target loss compared to the best offline performance within the surrogate loss framework. Since Van der Hoeven (2020) explicitly introduced the surrogate regret in online classification, follow-up studies have adopted this measure and extended it to structured prediction (Van der Hoeven et al., 2021; Sakaue et al., 2024; Shibukawa et al., 2025). Notably, this line of work has achieved finite surrogate regret bounds, which ensure that $R_T$ is upper bounded independently of $T$, under full-information feedback (i.e., $y_t \in \mathcal{Y}$ is revealed). This is an elegant extension of the classical finite mistake bound of the perceptron for linearly separable binary classification (Rosenblatt, 1958): under separability, the hinge loss $L(\boldsymbol{U}\boldsymbol{x}_t, y_t)$ of a suitable comparator $\boldsymbol{U}$ is zero (e.g., Orabona, 2025, Section 8.2), so a finite bound on $R_T$ translates into a finite mistake bound. In the prior literature, such finite bounds have been obtained by combining the standard (static) regret analysis of online convex optimization (OCO) for surrogate losses with the technique of *exploiting the surrogate gap* (Van der Hoeven, 2020), which we review in Section 3.1.

However, finite surrogate regret bounds fall short in non-stationary environments. To illustrate with an intuitive example, consider an online binary classification instance that is linearly separable for the first $T/2$ rounds, but then all labels are flipped for the remaining $T/2$ rounds. Figure 1 illustrates this setting. In this scenario, no fixed offline estimator $\boldsymbol{U}$ can avoid incurring a cumulative surrogate loss of $\sum_{t=1}^{T} L(\boldsymbol{U}\boldsymbol{x}_t, y_t) = \Omega(T)$. Consequently, even though $R_T$ is finite, the resulting upper bound on the cumulative target loss $\sum_{t=1}^{T} \ell(\hat{y}_t, y_t)$ grows linearly with $T$, making the guarantee meaningless. Is it then possible to obtain upper bounds on the cumulative target loss that do not grow linearly with $T$ even in non-stationary settings?

### 1.1. Our Contribution

We focus on full-information online structured prediction and establish the following bound on the target loss: for any comparator sequence $\boldsymbol{U}_1, \ldots, \boldsymbol{U}_T$, it holds that

$$\sum_{t=1}^{T} \ell(\hat{y}_t, y_t) = F_T + O(1 + P_T), \qquad (2)$$

where

$$F_T = \sum_{t=1}^{T} L(\boldsymbol{U}_t \boldsymbol{x}_t, y_t), \quad P_T = \sum_{t=2}^{T} \|\boldsymbol{U}_t - \boldsymbol{U}_{t-1}\|_{\mathrm{F}},$$

and $\|\cdot\|_{\mathrm{F}}$ denotes the Frobenius norm. In non-stationary environments, this bound can offer significantly stronger

---

[2] We write $\sum_{t=1}^{T} L(\boldsymbol{U}\boldsymbol{x}_t, y_t)$ on the right-hand side to clarify that it is not a proper regret, following Van der Hoeven et al. (2021).

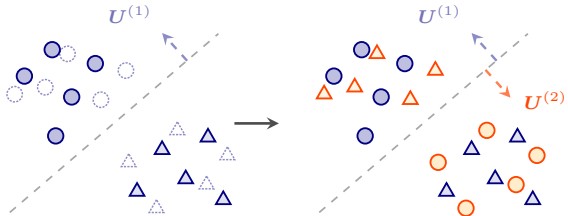

Figure 1. A simple non-stationary binary-classification instance. Circles and triangles represent labels $+1$ and $-1$, respectively. Solid blue markers indicate first-half examples; dashed blue markers in the left panel indicate examples that arrive in the second half, shown with their pre-flip labels. The same examples appear as solid orange markers with flipped labels in the right panel. The pre-flip labels are separable by $\boldsymbol{U}^{(1)}$, and the flipped labels are separable by $\boldsymbol{U}^{(2)} = -\boldsymbol{U}^{(1)}$, but the two halves taken together cannot be separated by any single fixed comparator.

guarantees than previous bounds of the form (1). For example, in the above non-stationary binary classification instance illustrated in Figure 1, we can set $\boldsymbol{U}_t$ to the best offline linear estimator $\boldsymbol{U}^{(1)}$ for the first $T/2$ rounds and to its negative $\boldsymbol{U}^{(2)}$ for the latter $T/2$ rounds, which yields $F_T = 0$ for surrogate losses with a separation margin (e.g., the smooth hinge loss) and $P_T = O(1)$, thus achieving $\sum_{t=1}^{T} \ell(\hat{y}_t, y_t) = O(1)$. This is far better than $\sum_{t=1}^{T} \ell(\hat{y}_t, y_t) = \Omega(T)$ implied by the existing finite bounds on $R_T$ in (1). Regret bounds that depend on $F_T$ are called *small-loss* bounds (Cesa-Bianchi and Lugosi, 2006; Zhao et al., 2024), and the quantity $P_T$ is called the *path length* (Zinkevich, 2003). Thus, we call a bound of the form (2) a "small-surrogate-loss + path-length" bound. Note that our bound (2) generalizes the finite bound on $R_T$ in (1) since setting $\boldsymbol{U}_1, \ldots, \boldsymbol{U}_T$ to the best offline $\boldsymbol{U}$ yields $P_T = 0$.

Similar to prior work (Van der Hoeven, 2020; Van der Hoeven et al., 2021; Sakaue et al., 2024; Shibukawa et al., 2025), our approach consists of two parts: OCO for surrogate losses and decoding of estimated score vectors. We use online gradient descent (OGD) for the OCO part, while we employ decoding methods of the previous studies. The key technical step in our analysis is a simple yet non-trivial learning-rate control of OGD. It combines the dynamic regret analysis of OGD, which yields the OCO regret bound $O(\sqrt{T}(1 + P_T))$ (Zinkevich, 2003; Hall and Willett, 2015), with the technique of exploiting the surrogate gap (see Section 3). This viewpoint motivates a Polyak-style learning rate, enabling a systematic derivation of target-loss bounds and delivering favorable empirical performance under non-stationarity (see Appendix E).

In Section 4, we tackle a significantly broader range of structured prediction tasks beyond the scope of prior work, including label ranking with the normalized discounted cumulative gain (NDCG) loss, where the label and prediction sets differ (see Appendix A.3). We achieve this using the

*convolutional Fenchel–Young loss*, a recent surrogate loss framework proposed by Cao et al. (2025). This framework is useful as it provides a principled way to construct desirable smooth surrogate losses based on target-loss structures. We introduce new technical tools for convolutional Fenchel–Young losses, which, combined with our main strategy in Section 3, enable us to establish the bound in (2) for more general structured prediction settings.

Finally, Section 5 presents a lower bound to show that the dependence on $F_T$ and $P_T$ in (2) is tight. This yields an interesting side observation: in online structured prediction, the classical approach of Zinkevich (2003) already suffices to establish tightness in $F_T$ and $P_T$. Recent work offers refined dynamic regret bounds in OCO (Zhang et al., 2018; Zhao et al., 2020, 2024), but these are not needed for our purpose.

To summarize, our main contributions are as follows:

- We establish the "small-surrogate-loss + path-length" bound (2) by synthesizing the dynamic regret analysis of OGD with the technique of exploiting the surrogate gap. Our analysis also sheds light on a Polyak-style learning rate with favorable empirical performance under non-stationarity (Section 3).

- We extend our approach to a broader class of structured prediction tasks through convolutional Fenchel–Young losses, enabled by new technical tools (Section 4).

- We prove a matching lower bound showing that (2) is tight in its dependence on $F_T$ and $P_T$ (Section 5).

### 1.2. Related Work

**Structured prediction.** We discuss particularly relevant studies on structured prediction; see, e.g., Sakaue et al. (2024, Appendix A) for a comprehensive overview. Our work is based on an inner-product representation of target losses, as detailed in Section 2.1. One such well-known framework is the *structure encoding loss function* (Ciliberto et al., 2016, 2020), and we adopt its extensions studied by Blondel (2019) and Cao et al. (2025). Regarding surrogate losses, our scope mainly covers, but is not limited to, *Fenchel–Young losses*, proposed by Blondel et al. (2020) (see Section 2.2). Also, our approach in Section 4 is based on the convolutional Fenchel–Young loss, proposed by Cao et al. (2025). This surrogate loss framework provides a useful family of smooth surrogate losses such that the surrogate excess risk linearly bounds the target excess risk from above (Cao et al., 2025, Theorem 15), a remarkable property in statistical learning. Note, however, that in our online setting, "small-surrogate-loss + path-length" bounds of the form (2) cannot be derived merely by using the convolutional Fenchel–Young loss as the surrogate; new technical lemmas introduced in Section 4.2 are required.

**Online classification and structured prediction.** Online classification has a broad literature; we refer the reader to an excellent overview by Van der Hoeven (2020, Section 1). Of particular relevance to our work, the classical perceptron (Rosenblatt, 1958) enjoys a finite mistake bound in the binary case under linear separability. Van der Hoeven (2020) extended this result to multiclass classification and obtained a finite surrogate regret bound (1), which holds regardless of separability. Sakaue et al. (2024) extended the result to online structured prediction with Fenchel–Young losses. Prior work on online classification/structured prediction also explored various limited feedback settings (Van der Hoeven, 2020; Van der Hoeven et al., 2021; Shibukawa et al., 2025), while our work, as the first step toward the non-stationary setting, focuses on full-information feedback. Addressing non-stationarity is crucial in online structured prediction. Indeed, Boudart et al. (2024) also studied non-stationary online structured prediction. This work, however, is not directly comparable to ours: they use a different performance metric (the standard regret in terms of the target loss), and their bounds explicitly involve $T$, unlike our bounds in (2).

**Dynamic regret bounds.** The (universal) dynamic regret in OCO compares the learner's choices with any time-varying comparators, thereby serving as a suitable performance measure in non-stationary environments. The path length $P_T$, introduced by Herbster and Warmuth (2001), is a common quantity representing the regularity of the comparator sequence. Zhang et al. (2018) achieved the optimal dynamic regret for OCO by showing upper and lower bounds with a rate of $\sqrt{T(1 + P_T)}$. Subsequent studies refined the bound by using data-dependent quantities, including the cumulative loss $F_T$ and the gradient variation (Zhao et al., 2020, 2024). These are significant in their own right, though the classical approach of Zinkevich (2003) is sufficient for our purpose, as noted in Section 1.1. Regarding the connection to online classification, Van der Hoeven et al. (2021) mentioned the idea of combining their method with OCO algorithms that enjoy dynamic regret bounds. However, this mention was only made as one example of how their proposed method can be potentially combined with various OCO algorithms, and no resulting bounds on the cumulative target loss were given.

**Polyak learning rates.** Polyak learning rates and their decreasing variants have been used in stochastic optimization (Loizou et al., 2021; Orvieto et al., 2022; Jiang and Stich, 2023). Our Polyak-style learning rate in Section 3.4 takes a similar form but serves a different purpose: translating the target–surrogate relationship into target-loss guarantees.

## 2. Preliminaries

For $d \in \mathbb{Z}_{>0}$, let $[d] = \{1, \ldots, d\}$. Let $\triangle^d = \{ \boldsymbol{\mu} \in \mathbb{R}_{\geq 0}^d : \|\boldsymbol{\mu}\|_1 = 1 \}$ denote the probability simplex in $\mathbb{R}^d$. Let $\|\boldsymbol{W}\|_{\mathrm{F}} = \sqrt{\mathrm{tr}(\boldsymbol{W}^\top \boldsymbol{W})}$ denote the Frobenius norm for

any matrix $\boldsymbol{W}$. For $\Omega \colon \mathbb{R}^d \to \mathbb{R} \cup \{+\infty\}$, let $\Omega^*(\boldsymbol{\theta}) = \sup\{\langle \boldsymbol{\theta}, \boldsymbol{\mu} \rangle - \Omega(\boldsymbol{\mu}) : \boldsymbol{\mu} \in \mathbb{R}^d\}$ be its convex conjugate and let $\operatorname{dom}(\Omega) = \{\boldsymbol{\mu} \in \mathbb{R}^d : \Omega(\boldsymbol{\mu}) < +\infty\}$.

## 2.1. Problem Setting

We present our problem setting. Examples of problems that fall into our setting include multiclass/multilabel classification and label ranking (see Appendices A and B for details). More examples can be found in Blondel (2019, Appendix A) and Sakaue et al. (2024, Section 2.3 and Appendix C).

**Structured prediction.** Let $\mathcal{Y} = [K]$ and $\hat{\mathcal{Y}} = [N]$ represent finite sets of labels and predictions, respectively. The set sizes, $K$ and $N$, can be extremely large when the sets consist of structured objects. A target loss $\ell \colon \hat{\mathcal{Y}} \times \mathcal{Y} \to \mathbb{R}_{\geq 0}$ measures the discrepancy between a prediction $\hat{y} \in \hat{\mathcal{Y}}$ and a ground-truth label $y \in \mathcal{Y}$. We can equivalently rewrite $\ell(\hat{y}, y)$ as $\langle \mathbf{e}^y, \boldsymbol{\ell}(\hat{y}) \rangle$, where $\mathbf{e}^y$ is the $y$th standard basis vector of $\mathbb{R}^K$ and $\boldsymbol{\ell}(\hat{y}) \in \mathbb{R}^K$ is the loss vector whose $y$th component is $\ell(\hat{y}, y)$ for $y \in \mathcal{Y}$. While this representation applies to any target losses, the vector size $K$ may be very large. Fortunately, more efficient lower-dimensional representations are available for many target losses. Here, we adopt the following $(\boldsymbol{\rho}, \boldsymbol{\ell^\rho})$-*decomposition* framework of Cao et al. (2025) to represent target losses efficiently.

**Definition 2.1.** The $(\boldsymbol{\rho}, \boldsymbol{\ell^\rho})$-decomposition of a target loss $\ell \colon \hat{\mathcal{Y}} \times \mathcal{Y} \to \mathbb{R}_{\geq 0}$ is given as follows:

$$\ell(\hat{y}, y) = \langle \boldsymbol{\rho}(y), \boldsymbol{\ell^\rho}(\hat{y}) \rangle + c(y), \tag{3}$$

where $\boldsymbol{\rho} \colon \mathcal{Y} \to \mathbb{R}^d$ is a label encoding function into the $d$-dimensional Euclidean space, $\boldsymbol{\ell^\rho} \colon \hat{\mathcal{Y}} \to \mathbb{R}^d$ is the corresponding loss encoding function, and $c \colon \mathcal{Y} \to \mathbb{R}$ is the term independent of prediction $\hat{y}$. We also define $\mathcal{L}^\rho \in \mathbb{R}^{d \times N}$ as the matrix whose column corresponding to $\hat{y} \in \hat{\mathcal{Y}}$ is $\boldsymbol{\ell^\rho}(\hat{y})$.

For example, in multiclass classification with $K = d$ classes, we let $\boldsymbol{\rho}(y) = \mathbf{e}^y$, $\boldsymbol{\ell^\rho}(\hat{y}) = \mathbf{1} - \mathbf{e}^{\hat{y}}$, where $\mathbf{1}$ is the all-ones vector, and $c \equiv 0$ to represent the 0-1 target loss. In multilabel classification, a label consists of $d$ binary outcomes, hence $K = 2^d$. Still, the standard Hamming loss enjoys a $(\boldsymbol{\rho}, \boldsymbol{\ell^\rho})$-decomposition with the dimensionality of $d = \log_2 K$; see Appendix A for details. Throughout this paper, we suppose that the target loss is represented as in (3) with some $\boldsymbol{\rho}$, $\boldsymbol{\ell^\rho}$, and $c$. Note that this entails no loss of generality since $\boldsymbol{\rho}(y) = \mathbf{e}^y$, $\boldsymbol{\ell^\rho}(\hat{y}) = \boldsymbol{\ell}(\hat{y})$, and $c \equiv 0$ always provide a valid (but possibly inefficient) $(\boldsymbol{\rho}, \boldsymbol{\ell^\rho})$-decomposition with $d = K$.

**Online learning protocol.** Let $\mathcal{X}$ be the space of input vectors. For $t = 1, \ldots, T$, the environment picks an input $\boldsymbol{x}_t \in \mathcal{X}$ and ground-truth $y_t \in \mathcal{Y}$. The learner observes $\boldsymbol{x}_t$, makes prediction $\hat{y}_t \in \hat{\mathcal{Y}}$, and observes feedback $y_t$. The prediction quality is measured by the target loss $\ell(\hat{y}_t, y_t)$. The learner's goal is to minimize the cumulative target loss,

$\sum_{t=1}^{T} \ell(\hat{y}_t, y_t)$, over $T$ rounds. The environment may select $(\boldsymbol{x}_t, y_t)$ adversarially based on information of past rounds.

**Surrogate loss framework.** The surrogate loss framework consists of the space of score vectors $\mathbb{R}^d$, a surrogate loss function $L \colon \mathbb{R}^d \times \mathcal{Y} \to \mathbb{R}_{\geq 0}$, which is convex in the first argument, and a decoding function that converts a score vector $\boldsymbol{\theta} \in \mathbb{R}^d$ into prediction $\hat{y} \in \hat{\mathcal{Y}}$. The score vectors and the loss-encoding vectors both lie in the same ambient space $\mathbb{R}^d$. For convenience, let $\boldsymbol{\pi} \colon \mathbb{R}^d \to \triangle^N$ denote a decoding *distribution*, where $\triangle^N$ is the probability simplex over the prediction set $\hat{\mathcal{Y}}$, allowing for randomized decoding from $\boldsymbol{\theta} \in \mathbb{R}^d$ to $\hat{y} \in \hat{\mathcal{Y}}$. Let $\mathcal{W}$ be a closed convex set of linear estimators. At the $t$th round, the learner computes $\boldsymbol{\theta}_t = \boldsymbol{W}_t \boldsymbol{x}_t$ with the current linear estimator $\boldsymbol{W}_t \in \mathcal{W}$ and makes a prediction $\hat{y}_t$ via sampling from the decoding distribution $\boldsymbol{\pi}(\boldsymbol{\theta}_t)$. The surrogate loss, $L(\boldsymbol{\theta}_t, y_t)$, quantifies the discrepancy between the estimated score vector $\boldsymbol{\theta}_t = \boldsymbol{W}_t \boldsymbol{x}_t$ and the ground-truth label $y_t \in \mathcal{Y}$. For brevity, let $L_t \colon \boldsymbol{W} \mapsto L(\boldsymbol{W} \boldsymbol{x}_t, y_t)$ denote the surrogate loss of $\boldsymbol{W} \in \mathcal{W}$ at round $t$, which is convex due to the convexity of $\boldsymbol{\theta} \mapsto L(\boldsymbol{\theta}, y)$. Thus, the learner can use OCO algorithms to learn $\boldsymbol{W}_t$ for $t = 1, \ldots, T$.

**Assumptions.** Below is a list of basic assumptions we make throughout this paper.

**Assumption 2.2.** *The following conditions hold:*

1. **Target loss can take zero.** $\min_{\hat{y} \in \hat{\mathcal{Y}}} \ell(\hat{y}, y) = 0$ *for any ground-truth* $y \in \mathcal{Y}$.

2. **Bounded input vectors.** $\|\boldsymbol{x}_t\|_2 \leq 1$ *for* $t \in [T]$.

3. **Bounded domain.** *There exists* $D > 0$ *such that* $\|\boldsymbol{W} - \boldsymbol{W}'\|_{\mathrm{F}} \leq D$ *for every* $\boldsymbol{W}, \boldsymbol{W}' \in \mathcal{W}$.

The first condition holds without loss of generality by adjusting $c(y)$ in Definition 2.1. The second and third bounds are for simplicity.

## 2.2. Fenchel–Young Loss

The *Fenchel–Young loss* framework (Blondel et al., 2020) offers a convenient recipe for designing various surrogate losses through regularization functions.

**Definition 2.3.** Let $\Omega \colon \mathbb{R}^d \to \mathbb{R} \cup \{+\infty\}$ be a function with $\operatorname{dom}(\Omega) \supseteq \operatorname{conv}(\{\boldsymbol{\rho}(y) \in \mathbb{R}^d : y \in \mathcal{Y}\})$. The Fenchel–Young loss $L_\Omega \colon \operatorname{dom}(\Omega^*) \times \mathcal{Y} \to \mathbb{R}_{\geq 0}$ generated by $\Omega$ is defined as follows:

$$L_\Omega(\boldsymbol{\theta}, y) = \Omega^*(\boldsymbol{\theta}) + \Omega(\boldsymbol{\rho}(y)) - \langle \boldsymbol{\theta}, \boldsymbol{\rho}(y) \rangle.$$

Fenchel–Young losses are convex in $\boldsymbol{\theta}$, non-negative, and take zero if and only if $\boldsymbol{\rho}(y) \in \partial \Omega^*(\boldsymbol{\theta})$ (Blondel et al., 2020, Proposition 2). Examples of Fenchel–Young losses include the logistic loss, CRF loss (Lafferty et al., 2001), and SparseMAP loss (Niculae et al., 2018); see Blondel et al. (2020, Table 1) for more examples. These surrogate losses

are generated by strongly convex regularizers $\Omega$, ensuring that the loss functions are smooth (Kakade et al., 2009, Theorem 6). The class of surrogate losses we consider is not limited to Fenchel–Young losses of the above form. We can deal with other surrogate losses, such as the smooth hinge loss, as described in Section 3.

### 2.3. Dynamic Regret Analysis of OGD

We recall a standard dynamic regret bound for OGD with non-increasing learning rates, which we use to derive our main results. This is a known extension of Zinkevich (2003, Theorem 2); see, e.g., Hall and Willett (2015, Theorem 2). We give a proof in Appendix D for completeness.

**Proposition 2.4.** *Compute* $\boldsymbol{W}_1, \ldots, \boldsymbol{W}_T \in \mathcal{W}$ *by applying OGD with non-increasing learning rate* $\eta_t > 0$ *to convex loss functions* $L_t \colon \mathcal{W} \to \mathbb{R}$ *for* $t = 1, \ldots, T$; *i.e., set* $\boldsymbol{W}_1$ *to an arbitrary point in* $\mathcal{W}$ *and let*

$$\boldsymbol{W}_{t+1} \leftarrow \arg\min\{ \| \boldsymbol{W}_t - \eta_t \boldsymbol{G}_t - \boldsymbol{W} \|_{\mathrm{F}} : \boldsymbol{W} \in \mathcal{W} \}$$

*for* $t = 1, \ldots, T$, *where* $\boldsymbol{G}_t \in \partial L_t(\boldsymbol{W}_t)$. *If the diameter of* $\mathcal{W}$ *is at most* $D$ *as in Assumption 2.2, then, for any* $\boldsymbol{U}_1, \ldots, \boldsymbol{U}_T \in \mathcal{W}$ *and* $P_T = \sum_{t=2}^{T} \| \boldsymbol{U}_t - \boldsymbol{U}_{t-1} \|_{\mathrm{F}}$, *we have*

$$\sum_{t=1}^{T}(L_t(\boldsymbol{W}_t) - L_t(\boldsymbol{U}_t)) \le \frac{D}{\eta_T}\left(\frac{D}{2} + P_T\right) + \sum_{t=1}^{T}\frac{\eta_t}{2}\|\boldsymbol{G}_t\|_{\mathrm{F}}^2.$$

If $\boldsymbol{U}_t$ are fixed to the best offline $\boldsymbol{U}$, then setting $\eta_t \asymp 1/\sqrt{T}$ for $t = 1, \ldots, T$ recovers the well-known $O(\sqrt{T})$ regret bound of OGD (regarding $D$ and $\|\boldsymbol{G}_t\|_{\mathrm{F}}$ as constants).

# 3. "Small-Surrogate-Loss + Path-Length" Bound via Surrogate Gap

This section considers the setting of prior work (Van der Hoeven, 2020; Van der Hoeven et al., 2021; Sakaue et al., 2024) and describes our main strategy for addressing non-stationarity. Previous studies obtained finite surrogate regret bounds in (1) with a powerful technique known as *exploiting the surrogate gap*. To leverage this, we further restrict the class of target losses from Definition 2.1, which was also assumed in the literature.

**Assumption 3.1.** *The prediction set equals the label set, i.e.,* $\hat{\mathcal{Y}} = \mathcal{Y}$. *In addition, there exist* $\boldsymbol{V} \in \mathbb{R}^{d \times d}$, $\boldsymbol{b} \in \mathbb{R}^d$, *and* $\gamma, \nu > 0$ *such that the following conditions hold for some norm* $\|\cdot\|$:

1. $\ell(\hat{y}, y) = \langle \boldsymbol{\rho}(\hat{y}), \boldsymbol{V}\boldsymbol{\rho}(y) + \boldsymbol{b} \rangle + c(y)$ *for any* $y \in \mathcal{Y}$ *and* $\hat{y} \in \hat{\mathcal{Y}}$,

2. $\mathbb{E}_{\hat{y}\sim\boldsymbol{\pi}}[\ell(\hat{y}, y)] \le \gamma\|\mathbb{E}_{\hat{y}\sim\boldsymbol{\pi}}[\boldsymbol{\rho}(\hat{y})] - \boldsymbol{\rho}(y)\|$ *for any* $\boldsymbol{\pi} \in \triangle^N$ *and* $y \in \mathcal{Y}$, *and*

3. $\|\boldsymbol{\rho}(y) - \boldsymbol{\rho}(y')\| \ge \nu$ *for any* $y, y' \in \mathcal{Y}$ *with* $y \ne y'$.

The first condition means that the target loss is affine decomposable (Blondel, 2019, Section 5). The second one requires that the expected target loss over a distribution $\boldsymbol{\pi}$ is small when the mean prediction is close to ground-truth $y$. The third one means that distinct labels have distant encodings. These have played essential roles in exploiting the surrogate gap in the previous studies. Examples satisfying these conditions include multiclass and multilabel classification (see Appendix A), and more examples are given in Sakaue et al. (2024, Section 2.3 and Appendix C). Section 4 addresses more general settings that may not satisfy Assumption 3.1. For example, the label ranking problem (see Appendix A.3 for details) has $\hat{\mathcal{Y}} \ne \mathcal{Y}$ and hence falls outside Assumption 3.1.

### 3.1. Decoding Methods with Surrogate Gaps

The key to exploiting the surrogate gap lies in decoding, or how to convert a score vector, $\boldsymbol{\theta} = \boldsymbol{W}\boldsymbol{x}$, into a prediction $\hat{y} \in \hat{\mathcal{Y}}$. Using existing randomized decoding methods, we can ensure that the expected target loss of $\hat{y}$ is strictly smaller than the surrogate loss of $\boldsymbol{\theta}$. In light of these randomized decoding methods, we use the following condition throughout this section.

**Assumption 3.2** (Surrogate gap condition). *Let* $\ell \colon \hat{\mathcal{Y}} \times \mathcal{Y} \to \mathbb{R}_{\ge 0}$ *be a target loss,* $L \colon \mathbb{R}^d \times \mathcal{Y} \to \mathbb{R}_{\ge 0}$ *a surrogate loss, and* $\alpha \in (0, 1)$. *There exists a decoding distribution map* $\boldsymbol{\pi} \colon \mathbb{R}^d \to \triangle^N$ *such that, for every score vector* $\boldsymbol{\theta} \in \mathbb{R}^d$, *it holds that*

$$\mathbb{E}_{\hat{y}\sim\boldsymbol{\pi}(\boldsymbol{\theta})}[\ell(\hat{y}, y)] \le (1-\alpha)L(\boldsymbol{\theta}, y) \quad \text{for any } y \in \mathcal{Y}.$$

This means that the expected target loss, $\mathbb{E}_{\hat{y}\sim\boldsymbol{\pi}(\boldsymbol{\theta})}[\ell(\hat{y}, y)]$, is smaller than the surrogate loss, $L(\boldsymbol{\theta}, y)$, by the margin $\alpha L(\boldsymbol{\theta}, y)$. This margin, called the *surrogate gap*, has played a central role in deriving finite surrogate regret bounds in the previous studies. Below are examples of the existing decoding methods:

- For $K$-class classification with the 0-1 target loss $\ell(\hat{y}, y) = \mathbb{1}_{\hat{y} \ne y}$, which takes one if $\hat{y} \ne y$ and zero otherwise, if $L$ is the smooth hinge loss or logistic loss, randomized decoding methods of Van der Hoeven (2020) and Van der Hoeven et al. (2021) satisfy Assumption 3.2 with $\alpha = 1/K$.

- For structured prediction problems with Assumption 3.1, if $L$ is a Fenchel–Young loss generated by $\Omega$ that is $\lambda$-strongly convex ($\lambda > \frac{4\gamma}{\nu}$) with respect to the norm $\|\cdot\|$ introduced in Assumption 3.1,[3] then the

---

[3] In Sakaue et al. (2024), $\Omega$ is additionally assumed to be of Legendre-type to ensure the self-bounding property. However, this additional condition is not needed: the self-bounding property follows from the strong convexity of $\Omega$ via the quadratic lower bound for Fenchel–Young losses (Blondel et al., 2022, Proposition 3).

randomized decoding method of Sakaue et al. (2024) satisfies Assumption 3.2 with $\alpha = \frac{4\gamma}{\lambda\nu}$.

Our analysis below applies whenever Assumption 3.2 holds; Assumption 3.1 is only a sufficient condition that guarantees the existence of decoding methods with Assumption 3.2. Still, among the structured prediction tasks we are aware of, all those admitting such decoding methods also satisfy Assumption 3.1.

### 3.2. Self-Bounding Surrogate Losses

We also introduce the following *self-bounding* property of surrogate loss functions.

**Assumption 3.3** (Self-bounding property). *Let $L_t \colon \mathcal{W} \ni \boldsymbol{W} \mapsto L(\boldsymbol{W}\boldsymbol{x}_t, y_t)$ be the surrogate loss at round $t \in [T]$. There exists $M > 0$ such that, for every $t \in [T]$ and $\boldsymbol{W} \in \mathcal{W}$, we can take a subgradient $\boldsymbol{G}_t(\boldsymbol{W}) \in \partial L_t(\boldsymbol{W})$ with*

$$\|\boldsymbol{G}_t(\boldsymbol{W})\|_{\mathrm{F}}^2 \le 2ML_t(\boldsymbol{W}).$$

This property was also assumed in the line of prior work; in particular, Van der Hoeven et al. (2021) has extensively discussed it.[4] The property holds for every non-negative $M$-smooth surrogate loss (Orabona, 2025, Theorem 4.24); hence, Fenchel–Young losses generated by $\frac{1}{M}$-strongly convex $\Omega$ satisfy it. Moreover, it holds for a broader class of losses beyond smooth ones. For example, the smooth hinge loss, viewed as a function of $\boldsymbol{W} \in \mathcal{W}$, is non-smooth (despite its name); still, it enjoys the self-bounding property, as detailed in Appendix B.1.[5]

### 3.3. Main Result

We present our main result based on the surrogate gap. The following bound has a significant advantage in non-stationary settings compared to the existing finite surrogate regret bounds (1). The proof is short once Proposition 2.4 is in place, although the bound itself is not obvious a priori.

**Theorem 3.4.** *Let $\alpha \in (0,1)$ be the surrogate-gap parameter given in Assumption 3.2 and $M > 0$ the self-bounding parameter given in Assumption 3.3. For $t = 1, \dots, T$, compute $\boldsymbol{W}_t \in \mathcal{W}$ using OGD with non-increasing learning rate $\eta_t$ such that*

$$\frac{\alpha}{M} \le \eta_t \le \frac{2\big(L_t(\boldsymbol{W}_t) - \mathbb{E}_{\hat{y}_t \sim \boldsymbol{\pi}(\boldsymbol{\theta}_t)}[\ell(\hat{y}_t, y_t)]\big)}{\|\boldsymbol{G}_t\|_{\mathrm{F}}^2}, \quad (4)$$

---

[4]Van der Hoeven et al. (2021) call self-bounding surrogate losses with an additional property *regular* surrogate losses. The terminology of "self-bounding" follows, e.g., Srebro et al. (2010) and Zhao et al. (2024).

[5]In Van der Hoeven et al. (2021, Appendix A), the hinge loss is also introduced as a self-bounding surrogate loss. However, their hinge loss is discontinuous and hence non-convex. We discuss this point in detail in Appendix C.

*where $\boldsymbol{\pi} \colon \mathbb{R}^d \to \triangle^N$ is a decoding distribution that satisfies Assumption 3.2 and $\boldsymbol{\theta}_t = \boldsymbol{W}_t \boldsymbol{x}_t$. The upper bound in (4) is interpreted as $+\infty$ when $\|\boldsymbol{G}_t\|_{\mathrm{F}} = 0$. The range of $\eta_t$ in (4) is non-empty, and, for any $\boldsymbol{U}_1, \dots, \boldsymbol{U}_T \in \mathcal{W}$, $P_T = \sum_{t=2}^{T}\|\boldsymbol{U}_t - \boldsymbol{U}_{t-1}\|_{\mathrm{F}}$, and $F_T = \sum_{t=1}^{T} L_t(\boldsymbol{U}_t)$, we have*

$$\sum_{t=1}^{T} \mathbb{E}_{\hat{y}_t \sim \boldsymbol{\pi}(\boldsymbol{\theta}_t)}[\ell(\hat{y}_t, y_t)] \le F_T + \frac{MD}{\alpha}\left(\frac{D}{2} + P_T\right).$$

*Proof.* First, we show that $\eta_t$ falling within the range in (4) always exists. From Assumptions 3.2 and 3.3, it holds that $L_t(\boldsymbol{W}_t) - \mathbb{E}_{\hat{y}_t \sim \boldsymbol{\pi}(\boldsymbol{\theta}_t)}[\ell(\hat{y}_t, y_t)] \ge \alpha L_t(\boldsymbol{W}_t) \ge \frac{\alpha}{2M}\|\boldsymbol{G}_t\|_{\mathrm{F}}^2$. Thus, setting $\eta_t$ to $\frac{\alpha}{M}$ always satisfies (4). To prove the main claim, we use Proposition 2.4 with $\eta_T \ge \frac{\alpha}{M}$ and (4) to obtain

$$\sum_{t=1}^{T} L_t(\boldsymbol{W}_t) - \sum_{t=1}^{T} L_t(\boldsymbol{U}_t)$$

$$\le \frac{MD}{\alpha}\left(\frac{D}{2} + P_T\right) + \sum_{t=1}^{T}\left(L_t(\boldsymbol{W}_t) - \mathbb{E}_{\hat{y}_t \sim \boldsymbol{\pi}(\boldsymbol{\theta}_t)}[\ell(\hat{y}_t, y_t)]\right).$$

Rearranging terms yields $\sum_{t=1}^{T} \mathbb{E}_{\hat{y}_t \sim \boldsymbol{\pi}(\boldsymbol{\theta}_t)}[\ell(\hat{y}_t, y_t)] \le F_T + \frac{MD}{\alpha}\left(\frac{D}{2} + P_T\right)$, as desired. $\square$

As discussed in Section 1, if the sequence $(\boldsymbol{x}_t, y_t)_{t=1}^{T}$ consists of $O(1)$ intervals of separable data, we can achieve $F_T = 0$ and $P_T = O(1)$ in Theorem 3.4 by changing $\boldsymbol{U}_t$ $O(1)$ times. This leads to an $O(1)$ bound on the cumulative target loss in this non-stationary setting, while the existing surrogate regret bound in (1) breaks down.

The essence of the proof lies in the design of the learning rate $\eta_t$ in (4). Specifically, the right-hand inequality in (4) enables us to cancel the cumulative surrogate loss terms, $\sum_{t=1}^{T} L_t(\boldsymbol{W}_t)$, and to derive an upper bound on the cumulative target loss. Meanwhile, Assumptions 3.2 and 3.3, corresponding to the surrogate-gap condition and the self-bounding property, respectively, ensure that the range of $\eta_t$ in (4) is non-empty; this implies $\eta_T \ge \frac{\alpha}{M}$ and thus prevents the $O((1 + P_T)/\eta_T)$ term in the OGD regret bound from becoming excessively large. This proof strategy extends the original idea of Van der Hoeven (2020) to the non-stationary setting and offers a more direct derivation of the target-loss bound through learning-rate adjustment in OGD. In the next subsection, we discuss how to choose a learning rate $\eta_t$ that satisfies (4).

As a side note, in the last part of the proof, the bound over the entire interval $t = 1, \dots, T$ is obtained simply by summing over $t = 1, \dots, T$. The same argument applies to any contiguous subinterval of $\{1, \dots, T\}$, with $F_T$ and $P_T$ defined on that subinterval. As such, our method enjoys an interval-wise guarantee akin to *strongly adaptive* bounds (Daniely et al., 2015), which are also desirable under non-stationarity.

## 3.4. Polyak-Style Learning Rate

The proof of Theorem 3.4 implies that, in theory, setting $\eta_t$ to the constant $\frac{\alpha}{M}$ is sufficient. This constant learning rate was employed by Van der Hoeven (2020) and Sakaue et al. (2024) (whereas Van der Hoeven et al. (2021) employed OCO methods with AdaGrad-type guarantees). In practice, however, $\eta_t = \frac{\alpha}{M}$ would be too conservative, resulting in slow adaptation to dynamic environments. Thus, we suggest a more adaptive non-increasing learning rate satisfying (4):

$$\eta_t = \min\left\{ \frac{2\big(L_t(\boldsymbol{W}_t) - \mathbb{E}_{\hat{y}_t \sim \boldsymbol{\pi}(\boldsymbol{\theta}_t)}[\ell(\hat{y}_t, y_t)]\big)}{\|\boldsymbol{G}_t\|_{\mathrm{F}}^2}, \eta_{t-1} \right\}, \quad (5)$$

where $\eta_0$ is any finite constant satisfying $\eta_0 \geq \alpha/M$. If $\|\boldsymbol{G}_t\|_{\mathrm{F}} = 0$, we interpret the first term in the minimum as $+\infty$, so that $\eta_t = \eta_{t-1}$, and make no update. Interestingly, this learning rate is analogous to Polyak's learning rate (Polyak, 1987), $\frac{L_t(\boldsymbol{W}_t) - \min_{\boldsymbol{W} \in \mathcal{W}} L_t(\boldsymbol{W})}{\|\boldsymbol{G}_t\|_{\mathrm{F}}^2}$, which is designed to minimize an upper bound on the suboptimality with respect to $L_t$. In our setting, the goal is to bound the expected target loss. This observation motivates replacing $\min_{\boldsymbol{W} \in \mathcal{W}} L_t(\boldsymbol{W})$ with $\mathbb{E}_{\hat{y}_t \sim \boldsymbol{\pi}(\boldsymbol{\theta}_t)}[\ell(\hat{y}_t, y_t)]$ (with an adjustment by a factor of two), as in (5). The minimum with $\eta_{t-1}$ is for ensuring the non-increasing property.[6] In Appendix E, we provide empirical evidence supporting the effectiveness of this Polyak-style learning rate; it matches or outperforms both the constant learning rate $\eta_t = \frac{\alpha}{M}$ and the AdaGrad-type learning rate, with its advantage growing as non-stationarity increases.

When computing the Polyak-style learning rate, the dominant additional quantity is $\mathbb{E}_{\hat{y}_t \sim \boldsymbol{\pi}(\boldsymbol{\theta}_t)}[\ell(\hat{y}_t, y_t)]$, an expectation over $\boldsymbol{\pi}(\boldsymbol{\theta}_t) \in \triangle^N$ with possibly exponential $N = |\hat{\mathcal{Y}}|$. This expectation can still be evaluated without enumerating $\hat{\mathcal{Y}}$: the randomized decoding of Sakaue et al. (2024) obtains $\boldsymbol{\pi}(\boldsymbol{\theta}_t)$ via Frank–Wolfe-type algorithms (e.g., Lacoste-Julien and Jaggi, 2015; Garber and Wolf, 2021) over the convex hull $\mathrm{conv}(\{\boldsymbol{\rho}(\hat{y}) \in \mathbb{R}^d : \hat{y} \in \hat{\mathcal{Y}}\})$, where each iteration requires only a linear optimization oracle over this convex hull. As discussed in Sakaue et al. (2024, Section 3.1), an $\varepsilon$-approximate decoding distribution can typically be computed with $O(d^2 \log(d/\varepsilon))$ such oracle calls; combined with active-set control (Beck and Shtern, 2017; Besançon et al., 2025), its support size is $O(d)$ as implied by Carathéodory's theorem. Thus, the expectation requires only $O(d)$ evaluations of $\ell(\cdot, y_t)$, and the per-round overhead is dominated, up to logarithmic factors, by $\widetilde{O}(d^2)$ linear optimization oracle calls. The same idea works with the decoding method for the convolutional Fenchel–Young loss discussed below.

---

[6]As noted in Section 1.2, decreasing Polyak learning rates are also used in stochastic optimization (Loizou et al., 2021; Orvieto et al., 2022; Jiang and Stich, 2023), but serve a different purpose.

## 4. Addressing More General Settings

We address more general structured prediction settings that may not satisfy the additional conditions in Assumption 3.1, thereby covering a broader range of tasks than prior work (Sakaue et al., 2024). In fact, many important problems, including label ranking with the normalized discounted cumulative gain (NDCG) loss (see Appendix A.3 for details) and precision at $k$ (Blondel, 2019, Appendix A), do not satisfy $\hat{\mathcal{Y}} = \mathcal{Y}$ and therefore fall outside Assumption 3.1. We tackle such general settings by leveraging the convolutional Fenchel–Young loss framework.

### 4.1. Convolutional Fenchel–Young Loss

The convolutional Fenchel–Young loss, proposed by Cao et al. (2025), provides an appealing framework for designing smooth surrogate losses that reflect structures of target losses and label sets. Below is the definition.

**Definition 4.1** (Convolutional Fenchel–Young loss). Given a $(\boldsymbol{\rho}, \boldsymbol{\ell^\rho})$-decomposition of the target loss $\ell$ in Definition 2.1, define $\tau \colon \mathbb{R}^d \to \mathbb{R}$ as follows:[7]

$$\tau(\boldsymbol{\mu}) = -\min\big\{ \langle \boldsymbol{\mu}, \boldsymbol{\ell^\rho}(\hat{y}) \rangle : \hat{y} \in \hat{\mathcal{Y}} \big\}.$$

Then, the convolutional Fenchel–Young loss $L_{\Omega_\tau} \colon \mathbb{R}^d \times \mathcal{Y} \to \mathbb{R}_{\geq 0}$ is defined as follows:

$$L_{\Omega_\tau}(\boldsymbol{\theta}, y) = \Omega_\tau^*(\boldsymbol{\theta}) + \Omega_\tau(\boldsymbol{\rho}(y)) - \langle \boldsymbol{\theta}, \boldsymbol{\rho}(y) \rangle,$$

where $\Omega_\tau(\boldsymbol{\mu}) = \Omega(\boldsymbol{\mu}) + \tau(\boldsymbol{\mu})$ for any $\boldsymbol{\mu} \in \mathbb{R}^d$ and $\Omega \colon \mathbb{R}^d \to \mathbb{R} \cup \{+\infty\}$ is a regularization function.

A convolutional Fenchel–Young loss is merely a specific instance of Fenchel–Young losses with regularizer $\Omega_\tau$. Nevertheless, this particular design is noteworthy in statistical learning because, when $\Omega$ is strongly convex, it provides a smooth surrogate loss such that the target excess risk is bounded linearly in the surrogate excess risk (Cao et al., 2025, Theorem 15).

Below, let $\mathcal{C} = \mathrm{conv}\big(\{\boldsymbol{\rho}(y) \in \mathbb{R}^d : y \in \mathcal{Y}\}\big)$ and assume that $\Omega$ in Definition 4.1 satisfies the following condition.[8]

**Assumption 4.2.** The function $\Omega$ in Definition 4.1 satisfies $\mathrm{dom}(\Omega) = \mathcal{C}$ and is $\lambda$-strongly convex over $\mathcal{C}$ with respect to the $\ell_2$-norm for some $\lambda > 0$.

This is a mild requirement. We can always satisfy it by defining $\Omega$ as the sum of a $\lambda$-strongly convex function $\Psi$ such that $\mathcal{C} \subseteq \mathrm{dom}(\Psi)$ and the indicator function of $\mathcal{C}$.

---

[7]This represents the negative of the Bayes risk of $\ell$ at $\boldsymbol{\eta} \in \triangle^K$ with $\boldsymbol{\mu} = \mathbb{E}_{y \sim \boldsymbol{\eta}}[\boldsymbol{\rho}(y)]$, shifted by $\mathbb{E}_{y \sim \boldsymbol{\eta}}[c(y)]$.

[8]Under Assumption 4.2, $\Omega$ is a closed proper convex function and $\mathrm{dom}(\Omega)$ is bounded. Consequently, $\Omega$ is *co-finite* and hence $\mathrm{dom}(\Omega^*) = \mathbb{R}^d$ holds (Rockafellar, 1970, Corollary 13.3.1). These guarantee that Cao et al. (2025, Condition 1) holds.

An example of convolutional Fenchel–Young losses with $\Omega$ being the negative Shannon entropy is given in Cao et al. (2025, Section 4). In addition, Appendix B.3 discusses an example for label ranking with the NDCG target loss (see also Appendix A.3). Blondel et al. (2020, Section 3.2) also discuss regularizers satisfying Assumption 4.2.

### 4.2. Key Lemmas

While the convolutional Fenchel–Young loss has been thoroughly investigated by Cao et al. (2025) from the perspective of statistical learning, applying it to our online learning setting requires new technical tools. The proofs of the following lemmas are given in Appendix F.

One such tool is the following target–surrogate relationship, which serves as an alternative to the surrogate-gap condition (Assumption 3.2) in Section 3. While this is a newly established relationship, it is related to Cao et al. (2025, Theorem 15); see Appendix F.2 for details.

**Lemma 4.3** (Target–surrogate relation). *Let $(\boldsymbol{\theta}, y) \in \mathbb{R}^d \times \mathcal{Y}$ and define a decoding distribution as*

$$\boldsymbol{\pi}(\boldsymbol{\theta}) \in \arg\min_{\boldsymbol{\pi} \in \triangle^N} \Omega^*(\boldsymbol{\theta} + \mathcal{L}^{\boldsymbol{\rho}} \boldsymbol{\pi}). \qquad (6)$$

*Draw $\hat{y}$ from $\hat{\mathcal{Y}}$ following $\boldsymbol{\pi}(\boldsymbol{\theta}) \in \triangle^N$. Then, we have*

$$\mathbb{E}_{\hat{y} \sim \boldsymbol{\pi}(\boldsymbol{\theta})}[\ell(\hat{y}, y)] = L_{\Omega_\tau}(\boldsymbol{\theta}, y) - L_\Omega(\boldsymbol{\theta} + \mathcal{L}^{\boldsymbol{\rho}} \boldsymbol{\pi}(\boldsymbol{\theta}), y). \qquad (7)$$

The distribution $\boldsymbol{\pi}(\boldsymbol{\theta})$ in (6) can be computed efficiently in most cases, with the support size being polynomial in $d$; see Cao et al. (2025, Section 4) and Appendix B.3.

Moreover, the following lower bound on $L_\Omega(\boldsymbol{\theta} + \mathcal{L}^{\boldsymbol{\rho}} \boldsymbol{\pi}(\boldsymbol{\theta}), y)$, the subtracted term in (7), serves as an alternative to the self-bounding property (Assumption 3.3).

**Lemma 4.4.** *Under Assumption 4.2 and the same conditions as Lemma 4.3, it holds that*

$$L_\Omega(\boldsymbol{\theta} + \mathcal{L}^{\boldsymbol{\rho}} \boldsymbol{\pi}(\boldsymbol{\theta}), y) \geq \frac{\lambda}{2} \|\nabla L_{\Omega_\tau}(\boldsymbol{\theta}, y)\|_2^2,$$

*where the convolutional Fenchel–Young loss is differentiable in its first argument (Cao et al., 2025, Corollary 11), and hence the gradient $\nabla L_{\Omega_\tau}(\boldsymbol{\theta}, y)$ is well defined.*

Intuitively, this lower bound is a quadratic-growth consequence of the strong convexity of $\Omega$, combined with the envelope theorem used in the analysis of the convolutional Fenchel–Young loss.

### 4.3. Main Result

We are ready to prove our main result, a "small-surrogate-loss + path-length" bound for the broader class of tasks that may not satisfy Assumption 3.1. The following theorem is

obtained by applying the OGD scheme used in Theorem 3.4 to the convolutional Fenchel–Young loss. Thus, we can use the Polyak-style learning rate given in Section 3.4.

**Theorem 4.5.** *Let $L_t \colon \boldsymbol{W} \mapsto L_{\Omega_\tau}(\boldsymbol{W} \boldsymbol{x}_t, y_t)$ be a convolutional Fenchel–Young loss, where $\Omega_\tau = \Omega + \tau$ and $\Omega$ satisfies Assumption 4.2. For $t = 1, \ldots, T$, compute $\boldsymbol{W}_t \in \mathcal{W}$ using OGD with non-increasing learning rate $\eta_t$ such that*

$$\lambda \leq \eta_t \leq \frac{2\big(L_t(\boldsymbol{W}_t) - \mathbb{E}_{\hat{y}_t \sim \boldsymbol{\pi}(\boldsymbol{\theta}_t)}[\ell(\hat{y}_t, y_t)]\big)}{\|\boldsymbol{G}_t\|_{\mathrm{F}}^2}, \qquad (8)$$

*where $\boldsymbol{\theta}_t = \boldsymbol{W}_t \boldsymbol{x}_t$ and $\boldsymbol{\pi}(\boldsymbol{\theta}_t) \in \arg\min_{\boldsymbol{\pi} \in \triangle^N} \Omega^*(\boldsymbol{\theta}_t + \mathcal{L}^{\boldsymbol{\rho}} \boldsymbol{\pi})$ is the decoding distribution given in (6). As in Theorem 3.4, the upper bound in (8) is interpreted as $+\infty$ when $\|\boldsymbol{G}_t\|_{\mathrm{F}} = 0$. The range of $\eta_t$ in (8) is non-empty and, for any $\boldsymbol{U}_1, \ldots, \boldsymbol{U}_T \in \mathcal{W}$, $P_T = \sum_{t=2}^{T} \|\boldsymbol{U}_t - \boldsymbol{U}_{t-1}\|_{\mathrm{F}}$, and $F_T = \sum_{t=1}^{T} L_t(\boldsymbol{U}_t)$, we have*

$$\sum_{t=1}^{T} \mathbb{E}_{\hat{y}_t \sim \boldsymbol{\pi}(\boldsymbol{\theta}_t)}[\ell(\hat{y}_t, y_t)] \leq F_T + \frac{D}{\lambda}\left(\frac{D}{2} + P_T\right).$$

*Proof.* Again, we begin by confirming that there exists $\eta_t$ that satisfies (8). Lemmas 4.3 and 4.4 imply

$$L_t(\boldsymbol{W}_t) - \mathbb{E}_{\hat{y}_t \sim \boldsymbol{\pi}(\boldsymbol{\theta}_t)}[\ell(\hat{y}_t, y_t)] = L_\Omega(\boldsymbol{\theta}_t + \mathcal{L}^{\boldsymbol{\rho}} \boldsymbol{\pi}(\boldsymbol{\theta}_t), y_t)$$

$$\geq \frac{\lambda}{2} \|\nabla L_{\Omega_\tau}(\boldsymbol{\theta}_t, y_t)\|_2^2,$$

and the right-hand side is bounded below by $\frac{\lambda}{2} \|\boldsymbol{G}_t\|_{\mathrm{F}}^2$ since $\|\boldsymbol{G}_t\|_{\mathrm{F}}^2 = \|\nabla L_{\Omega_\tau}(\boldsymbol{\theta}_t, y_t)\|_2^2 \|\boldsymbol{x}_t\|_2^2 \leq \|\nabla L_{\Omega_\tau}(\boldsymbol{\theta}_t, y_t)\|_2^2$ holds from $\|\boldsymbol{x}_t\|_2 \leq 1$ (Assumption 2.2). Therefore, $\eta_t = \lambda$ always satisfies (8). To prove the main claim, we use Proposition 2.4 with $\eta_T \geq \lambda$ and (8), obtaining

$$\sum_{t=1}^{T}(L_t(\boldsymbol{W}_t) - L_t(\boldsymbol{U}_t))$$

$$\leq \frac{D}{\lambda}\left(\frac{D}{2} + P_T\right) + \sum_{t=1}^{T}\left(L_t(\boldsymbol{W}_t) - \mathbb{E}_{\hat{y}_t \sim \boldsymbol{\pi}(\boldsymbol{\theta}_t)}[\ell(\hat{y}_t, y_t)]\right).$$

Rearranging terms yields $\sum_{t=1}^{T} \mathbb{E}_{\hat{y}_t \sim \boldsymbol{\pi}(\boldsymbol{\theta}_t)}[\ell(\hat{y}_t, y_t)] \leq F_T + \frac{D}{\lambda}\left(\frac{D}{2} + P_T\right)$, as desired. $\square$

Let us discuss the difference from the analysis in Theorem 3.4. In that proof, we used the surrogate-gap condition, $L_t(\boldsymbol{W}_t) - \mathbb{E}_{\hat{y}_t \sim \boldsymbol{\pi}(\boldsymbol{\theta}_t)}[\ell(\hat{y}_t, y_t)] \geq \alpha L_t(\boldsymbol{W}_t)$, to obtain the desired bound. In the general setting considered here, no decoding method is known that guarantees such a condition. Fortunately, however, Lemmas 4.3 and 4.4 enable us to obtain the alternative condition, $L_t(\boldsymbol{W}_t) - \mathbb{E}_{\hat{y}_t \sim \boldsymbol{\pi}(\boldsymbol{\theta}_t)}[\ell(\hat{y}_t, y_t)] \geq \frac{\lambda}{2} \|\boldsymbol{G}_t\|_{\mathrm{F}}^2$. This requirement is easier to satisfy in light of $\|\boldsymbol{G}_t\|_{\mathrm{F}}^2 \leq \frac{2}{\lambda} L_t(\boldsymbol{W}_t)$, which follows

from the self-bounding property of $L_t$ (Assumption 3.3) implied by the $\lambda$-strong convexity of $\Omega_\tau$. The above proof shows that this weaker requirement is sufficient to derive the "small-surrogate-loss + path-length" bound.

As a cautionary note, one might be tempted to increase the strong convexity parameter $\lambda$ arbitrarily to shrink the $O((1 + P_T)/\lambda)$ term; however, this also increases the scale of $L_{\Omega_\tau}$ and thus enlarges $F_T$ (see also Blondel et al., 2020, Proposition 2). Therefore, the value of $\lambda$ should be chosen to balance this trade-off.

## 5. Lower Bound

We present a lower bound that complements our "small-surrogate-loss + path-length" upper bounds discussed so far.

**Theorem 5.1.** *For any possibly randomized learner and any nonnegative integers $T_F, T_P$ with $T = T_F + T_P > 0$, there exists an instance of online binary classification with the 0-1 target loss and the smooth hinge surrogate loss, together with a comparator sequence $U_1, \ldots, U_T$, such that*

$$\mathbb{E}\left[\sum_{t=1}^T \ell(\hat{y}_t, y_t)\right] \geq \frac{T}{2}, \quad F_T \leq 3T_F, \quad \text{and} \quad P_T \leq \sqrt{2}T_P,$$

*where the expectation is taken over the learner's possible randomness. Consequently, for the same comparator sequence, the expected target loss is $\Omega(F_T + P_T)$. Equivalently, the learner can suffer $\Omega(T_F + T_P)$ expected target loss while there exist comparators with $F_T = O(T_F)$ and $P_T = O(T_P)$.*

We present the proof in Appendix G. Since our upper bounds hold for any comparator sequence, it suffices to identify comparator sequences for which the lower bound holds. This lower bound suggests that the joint linear dependence on $F_T$ and $P_T$ cannot be improved in the worst case.

Note that the lower bound might be avoided by focusing on specific instances other than binary classification with the smooth hinge surrogate loss. Theorem 5.1 is intended to complement our upper bounds, which apply to general settings that subsume this specific instance.

## 6. Conclusion

We have established "small-surrogate-loss + path-length" bounds on the cumulative target loss for non-stationary online structured prediction. The core idea is to synthesize the dynamic regret analysis of OGD with the surrogate gap. En route to this, we have introduced a Polyak-style learning rate, which directly yields target-loss guarantees and works well empirically. We have also extended this approach to a broader range of problem settings than previously considered by leveraging the convolutional Fenchel–Young loss.

This extension is enabled by new technical lemmas on the convolutional Fenchel–Young loss. Finally, we have obtained a lower bound to complement the upper bounds.

We close with limitations and future directions. This work is restricted to full-information feedback. Existing surrogate regret bounds under limited feedback, such as bandit feedback, explicitly depend on $T$ (Van der Hoeven, 2020; Van der Hoeven et al., 2021; Shibukawa et al., 2025), unlike their full-information counterparts. Obtaining meaningful dynamic-regret-based guarantees in such settings remains a challenging direction for future work. Furthermore, non-stationarity has also been studied from a stability-based perspective (Huang and Wang, 2025). Clarifying the relationship between this line of work and our "small-surrogate-loss + path-length" guarantees is another interesting direction.

## Acknowledgements

SS is supported by JST BOOST Program Grant Number JPMJBY24D1. HB is supported by JST PRESTO Grant Number JPMJPR24K6. YC is supported by Google PhD Fellowship program. The authors thank the anonymous reviewers for their helpful reviews and comments.

## Impact Statement

This paper presents work whose goal is to advance the field of Machine Learning. There are many potential societal consequences of our work, none of which we feel must be specifically highlighted here.

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

# A. Examples of Structured Prediction Problems

Below we discuss some examples provided in Blondel (2019) and Sakaue et al. (2024).

## A.1. Multiclass Classification

Let $\mathcal{Y} = \hat{\mathcal{Y}} = [K]$ and $\ell(\hat{y}, y) = \mathbb{1}_{\hat{y} \neq y}$ be the 0-1 loss, which trivially satisfies the zero-loss condition in Assumption 2.2. An affine decomposition of the dimensionality $d = K$ (see Assumption 3.1) is given by $\boldsymbol{\rho}(y) = \mathbf{e}^y$, $\boldsymbol{V} = \mathbf{1}\mathbf{1}^\top - I$, $\boldsymbol{b} = 0$, and $c \equiv 0$, where $\mathbf{1} \in \mathbb{R}^d$ is the all-ones vector and $I \in \mathbb{R}^{d \times d}$ is the identity matrix, i.e., $\ell(\hat{y}, y) = \langle \mathbf{e}^{\hat{y}}, (\mathbf{1}\mathbf{1}^\top - I)\mathbf{e}^y \rangle = \langle \mathbf{e}^{\hat{y}}, \mathbf{1} - \mathbf{e}^y \rangle$. Let $\boldsymbol{\pi} \in \triangle^d$ and $\hat{\boldsymbol{\mu}} = \mathbb{E}_{\hat{y} \sim \boldsymbol{\pi}} \mathbf{e}^{\hat{y}} \in \triangle^d$. Then, we have $\mathbb{E}_{\hat{y} \sim \boldsymbol{\pi}}[\ell(\hat{y}, y)] = \langle \hat{\boldsymbol{\mu}}, \mathbf{1} - \mathbf{e}^y \rangle = \frac{1}{2}\|\hat{\boldsymbol{\mu}} - \mathbf{e}^y\|_1$ for any $y \in \mathcal{Y}$ and $\|\mathbf{e}^y - \mathbf{e}^{y'}\|_1 = 2$ for any $y, y' \in \mathcal{Y}$ with $y \neq y'$. Therefore, the conditions in Assumption 3.1 are satisfied.

## A.2. Multilabel Classification

Consider predicting $d$ binary outcomes, 0 or 1. We identify $\mathcal{Y} = \hat{\mathcal{Y}}$ with $\{0, 1\}^d$ (equivalently, a set of size $K = 2^d$). For the target loss, we use the Hamming loss, $\ell(\hat{y}, y) = \frac{1}{d}\sum_{i=1}^d \mathbb{1}_{\hat{y}_i \neq y_i}$, where $\hat{y}_i$ and $y_i$ are the $i$th outcome of $\hat{y} \in \hat{\mathcal{Y}}$ and $y \in \mathcal{Y}$, respectively, for $i \in [d]$. This target loss trivially satisfies the zero-loss condition in Assumption 2.2. A $d$-dimensional affine decomposition of the Hamming loss is given as follows: let $\boldsymbol{\rho}(y) \in \{0, 1\}^d$ be a vector whose $i$th element represents the $i$th outcome of $y$, $\boldsymbol{V} = -\frac{2}{d}I$, $\boldsymbol{b} = \frac{1}{d}\mathbf{1}$, and $c(y) = \frac{1}{d}\langle \boldsymbol{\rho}(y), \mathbf{1} \rangle$; then, it holds that $\ell(\hat{y}, y) = \frac{1}{d}(\langle \boldsymbol{\rho}(\hat{y}), \mathbf{1} \rangle + \langle \boldsymbol{\rho}(y), \mathbf{1} \rangle - 2\langle \boldsymbol{\rho}(\hat{y}), \boldsymbol{\rho}(y) \rangle)$. We also have $\mathbb{E}_{\hat{y} \sim \boldsymbol{\pi}}[\ell(\hat{y}, y)] = \frac{1}{d}\|\mathbb{E}_{\hat{y} \sim \boldsymbol{\pi}} \boldsymbol{\rho}(\hat{y}) - \boldsymbol{\rho}(y)\|_1$ for any $y \in \mathcal{Y}$ and $\|\boldsymbol{\rho}(y) - \boldsymbol{\rho}(y')\|_1 \geq 1$ for any $y, y' \in \mathcal{Y}$ with $y \neq y'$. Therefore, Assumption 3.1 is satisfied.

## A.3. Ranking with NDCG Loss

Consider predicting a ranking of documents, a common task in information retrieval. Let $\mathcal{Y} = [k]^d$ be the set of relevance scores of $d$ documents and $\hat{\mathcal{Y}}$ the permutations of $[d]$. The normalized discounted cumulative gain (NDCG) loss $\ell \colon \hat{\mathcal{Y}} \times \mathcal{Y} \to \mathbb{R}_{\geq 0}$ with weights $w_1, \dots, w_d \geq 0$ is defined as follows:

$$\ell(\hat{y}, y) = 1 - \frac{1}{N(y)}\sum_{i=1}^d y_i w_{\hat{y}(i)},$$

where $y_i \in [k]$ is the relevance score of the $i$th document, $\hat{y}(i) \in [d]$ is the $i$th element of permutation $\hat{y}$ of $[d]$, and $N(y) = \max_{\hat{y} \in \hat{\mathcal{Y}}} \sum_{i=1}^d y_i w_{\hat{y}(i)}$ is the normalization constant; we assume $N(y) > 0$ for all $y \in \mathcal{Y}$. A $d$-dimensional $(\boldsymbol{\rho}, \boldsymbol{\ell}^{\boldsymbol{\rho}})$-decomposition of this loss is given by $\boldsymbol{\rho}(y) = -(y_1, \dots, y_d)^\top / N(y)$, $\boldsymbol{\ell}^{\boldsymbol{\rho}}(\hat{y}) = (w_{\hat{y}(1)}, \dots, w_{\hat{y}(d)})^\top$, and $c \equiv 1$. Setting $\hat{y}$ to the best permutation (a maximizer in the definition of $N(y)$) makes the loss zero. Note that this example does not satisfy the additional assumptions in Assumption 3.1 due to $\mathcal{Y} \neq \hat{\mathcal{Y}}$. Still, the framework in Section 4 can handle this case using a convolutional Fenchel–Young loss as a surrogate loss.

# B. Examples of Surrogate Losses

We provide examples of surrogate losses. All of them satisfy the self-bounding property in Assumption 3.3.

## B.1. Smooth Hinge Loss

We describe the details of the smooth hinge loss used in Van der Hoeven (2020) and Van der Hoeven et al. (2021). For any $\boldsymbol{W} \in \mathcal{W}$, we define the multiclass margin of $\boldsymbol{W}$ at round $t$ as follows:

$$m_t(\boldsymbol{W}, y_t) = \langle \mathbf{e}^{y_t}, \boldsymbol{W}\boldsymbol{x}_t \rangle - \max_{y \neq y_t}\langle \mathbf{e}^y, \boldsymbol{W}\boldsymbol{x}_t \rangle.$$

The smooth hinge loss of Rennie and Srebro (2005) (strictly speaking, its multiclass extension based on Crammer and Singer 2001) is defined as follows:

$$L_t(\boldsymbol{W}) = L(\boldsymbol{W}\boldsymbol{x}_t, y_t) = \begin{cases} 1 - 2m_t(\boldsymbol{W}, y_t) & \text{if } m_t(\boldsymbol{W}, y_t) \leq 0, \\ \max\{1 - m_t(\boldsymbol{W}, y_t), 0\}^2 & \text{if } m_t(\boldsymbol{W}, y_t) > 0. \end{cases}$$

We can check the convexity of $L_t(\boldsymbol{W})$ as follows: $m_t(\boldsymbol{W}, y_t)$ is concave in $\boldsymbol{W}$ since it is the negative of the pointwise maximum of linear functions, and $L_t(\boldsymbol{W})$ viewed as a univariate function of $m_t(\boldsymbol{W}, y_t)$ is convex and non-increasing; the composition of these two functions is convex (Boyd and Vandenberghe, 2004, Section 3.2.4).

For convenience, we also define

$$m_t^*(\boldsymbol{W}) = \max_{y \in \mathcal{Y}} m_t(\boldsymbol{W}, y) \qquad \text{and} \qquad y_t^* \in \arg\max_{y \in \mathcal{Y}} \langle \mathbf{e}^y, \boldsymbol{W} \boldsymbol{x}_t \rangle.$$

Note that, for any $y' \in \mathcal{Y} \setminus \{y_t^*\}$, we have

$$m_t(\boldsymbol{W}, y') = \langle \mathbf{e}^{y'}, \boldsymbol{W} \boldsymbol{x}_t \rangle - \max_{y \neq y'} \langle \mathbf{e}^y, \boldsymbol{W} \boldsymbol{x}_t \rangle = \langle \mathbf{e}^{y'}, \boldsymbol{W} \boldsymbol{x}_t \rangle - \langle \mathbf{e}^{y_t^*}, \boldsymbol{W} \boldsymbol{x}_t \rangle \leq 0.$$

In addition, if $y_t^* = y_t$, we have

$$m_t(\boldsymbol{W}, y_t) = \langle \mathbf{e}^{y_t^*}, \boldsymbol{W} \boldsymbol{x}_t \rangle - \max_{y \neq y_t} \langle \mathbf{e}^y, \boldsymbol{W} \boldsymbol{x}_t \rangle \geq 0.$$

Therefore, under $y_t^* = y_t$, we have $m_t^*(\boldsymbol{W}) = m_t(\boldsymbol{W}, y_t^*) = m_t(\boldsymbol{W}, y_t) \geq 0$.

Defining $\tilde{y}_t \in \arg\max_{y \neq y_t} \langle \mathbf{e}^y, \boldsymbol{W} \boldsymbol{x}_t \rangle$, we can express a subgradient of $L_t$ at $\boldsymbol{W}$ as follows:

$$\boldsymbol{G}_t(\boldsymbol{W}) = \begin{cases} 2(\mathbf{e}^{\tilde{y}_t} - \mathbf{e}^{y_t})\boldsymbol{x}_t^\top & \text{if } y_t^* \neq y_t, \\ 2(1 - m_t^*(\boldsymbol{W}))(\mathbf{e}^{\tilde{y}_t} - \mathbf{e}^{y_t})\boldsymbol{x}_t^\top & \text{if } y_t^* = y_t \text{ and } m_t^*(\boldsymbol{W}) < 1, \\ 0 & \text{if } y_t^* = y_t \text{ and } m_t^*(\boldsymbol{W}) \geq 1, \end{cases}$$

where $0$ means the all-zero matrix. Thus, $\|\boldsymbol{G}_t(\boldsymbol{W})\|_{\mathrm{F}} \leq 2\sqrt{2}\|\boldsymbol{x}_t\|_2 \leq 2\sqrt{2}$ holds in any case. As for the self-bounding property, the third case is trivial. In the first case, $y_t^* \neq y_t$ implies $m_t(\boldsymbol{W}, y_t) \leq 0$ and hence

$$\|\boldsymbol{G}_t(\boldsymbol{W})\|_{\mathrm{F}}^2 = 8\|\boldsymbol{x}_t\|_2^2 \leq 8L_t(\boldsymbol{W}).$$

In the second case, we have

$$\|\boldsymbol{G}_t(\boldsymbol{W})\|_{\mathrm{F}}^2 = 8(1 - m_t^*(\boldsymbol{W}))^2\|\boldsymbol{x}_t\|_2^2 \leq 8L_t(\boldsymbol{W}).$$

Thus, $\|\boldsymbol{G}_t(\boldsymbol{W})\|_{\mathrm{F}}^2 \leq 8L_t(\boldsymbol{W})$ holds in any case. Note that $L_t$ is non-smooth in $\boldsymbol{W} \in \mathcal{W}$ since, if multiple $\tilde{y}_t$ attain $\max_{y \neq y_t} \langle \mathbf{e}^y, \boldsymbol{W} \boldsymbol{x}_t \rangle$, then the subgradients are non-unique, violating smoothness (or Lipschitz continuity of the gradient). We remark that this non-smoothness is caused by the multiclass margin $m_t(\boldsymbol{W}, y_t)$; indeed, $L_t$ viewed as a univariate function of $m_t(\boldsymbol{W}, y_t)$ is smooth.

### B.2. Fenchel–Young Loss

Consider using the Fenchel–Young loss defined in Section 2.2 as a surrogate loss, namely,

$$L_t(\boldsymbol{W}) = L_\Omega(\boldsymbol{W} \boldsymbol{x}_t, y_t) = \Omega^*(\boldsymbol{W} \boldsymbol{x}_t) + \Omega(\boldsymbol{\rho}(y_t)) - \langle \boldsymbol{W} \boldsymbol{x}_t, \boldsymbol{\rho}(y_t) \rangle.$$

We assume the regularizer $\Omega$ to be $\lambda$-strongly convex with respect to the Euclidean norm. For convenience, let $\mathcal{C} = \text{conv}\left(\left\{ \boldsymbol{\rho}(y) \in \mathbb{R}^d : y \in \mathcal{Y} \right\}\right)$ and $\boldsymbol{\theta} = \boldsymbol{W} \boldsymbol{x}_t$. We define the regularized prediction (strictly speaking, its encoded vector) as

$$\boldsymbol{\rho}_\Omega(\boldsymbol{\theta}) = \arg\max\{ \langle \boldsymbol{\theta}, \boldsymbol{\mu} \rangle - \Omega(\boldsymbol{\mu}) : \boldsymbol{\mu} \in \mathcal{C} \}.$$

Then, it holds that $\nabla L_t(\boldsymbol{W}) = (\boldsymbol{\rho}_\Omega(\boldsymbol{\theta}) - \boldsymbol{\rho}(y_t))\boldsymbol{x}_t^\top$ (Blondel et al., 2020, Proposition 2), which means that $\|\nabla L_t(\boldsymbol{W})\|_{\mathrm{F}}$ is at most the $\ell_2$-diameter of $\mathcal{C}$. Moreover, we have $\frac{\lambda}{2}\|\boldsymbol{\rho}_\Omega(\boldsymbol{\theta}) - \boldsymbol{\rho}(y_t)\|_2^2 \leq L_t(\boldsymbol{W})$ due to the $\lambda$-strong convexity of $\Omega$ (Blondel et al., 2022, Proposition 3), and thus the self-bounding property holds with $M = 1/\lambda$. Below are two examples of the Fenchel–Young loss.

**Logistic loss.** The logistic loss can be seen as a Fenchel–Young loss. Consider the classification problem with $K$ classes. Let $d = K$ and $\boldsymbol{\rho}(y) = \mathbf{e}^y$ for $y \in \mathcal{Y} = [K]$ and $\mathcal{C} = \triangle^K$. If we adopt the negative Shannon entropy,

$\Omega(\boldsymbol{\mu}) = \sum_{y \in \mathcal{Y}} \mu_y \ln \mu_y$ for $\boldsymbol{\mu} \in \triangle^K$, as a regularizer, the resulting Fenchel–Young loss is the logistic loss (Blondel et al., 2020, Section 4.3):

$$L_t(\boldsymbol{W}) = L_\Omega(\boldsymbol{W}\boldsymbol{x}_t, y_t) = \ln \sum_{y \in \mathcal{Y}} \exp(\theta_y) - \theta_{y_t},$$

where $\boldsymbol{\theta} = \boldsymbol{W}\boldsymbol{x}_t$ and $\theta_y = \langle \mathbf{e}^y, \boldsymbol{W}\boldsymbol{x}_t \rangle$ for $y \in \mathcal{Y}$. In this case, the regularized prediction, $\boldsymbol{\rho}_\Omega$, equals the softmax function. The negative Shannon entropy is 1-strongly convex on $\triangle^K$ with respect to the $\ell_1$-norm, and hence with respect to the $\ell_2$-norm. Thus, the self-bounding property holds with $M = 1$. Note that in some cases the base of the logarithm in the logistic loss is not Euler's number e but rather 2 or $K$ (Van der Hoeven 2020; Van der Hoeven et al. 2021; Sakaue et al. 2024); accordingly, the Lipschitz constant and the value of $M$ also change.

**SparseMAP loss.** The SparseMAP loss, proposed by Niculae et al. (2018), is another example of a Fenchel–Young loss that applies to general structured prediction problems. Let $\mathcal{Y} = [K]$ be a finite set of labels and $\mathcal{C} = \operatorname{conv}\big(\{\boldsymbol{\rho}(y) \in \mathbb{R}^d : y \in \mathcal{Y}\}\big)$. Let

$$\Omega(\boldsymbol{\mu}) = \begin{cases} \frac{1}{2}\|\boldsymbol{\mu}\|_2^2 & \text{if } \boldsymbol{\mu} \in \mathcal{C}, \\ +\infty & \text{otherwise.} \end{cases}$$

The resulting Fenchel–Young loss, $\boldsymbol{\theta} \mapsto L_\Omega(\boldsymbol{\theta}, y)$, is called the SparseMAP loss, whose regularized prediction $\boldsymbol{\rho}_\Omega(\boldsymbol{\theta}) = \arg\max\{\langle \boldsymbol{\theta}, \boldsymbol{\mu} \rangle - \Omega(\boldsymbol{\mu}) : \boldsymbol{\mu} \in \mathcal{C}\}$ tends to have a sparse support. Since $\Omega$ is 1-strongly convex with respect to the $\ell_2$-norm on $\mathcal{C}$, $L_\Omega$ is 1-smooth and hence self-bounding with $M = 1$.

### B.3. Convolutional Fenchel–Young Loss: The Ranking Case

The convolutional Fenchel–Young loss defined in Definition 4.1 satisfies the self-bounding property similarly to Fenchel–Young losses. Below, we discuss the computational aspect when applying our method with the convolutional Fenchel–Young loss to the label ranking problem with the NDCG loss in Appendix A.3. Recall that the convolutional Fenchel–Young loss is an instance of the Fenchel–Young loss generated by $\Omega_\tau = \Omega + \tau$, where $\tau(\boldsymbol{\mu}) = -\min_{\hat{y} \in \hat{\mathcal{Y}}}\langle \boldsymbol{\mu}, \boldsymbol{\ell}^\rho(\hat{y}) \rangle$. As in Assumption 4.2, we assume that $\Omega$ is a $\lambda$-strongly convex function with $\operatorname{dom}(\Omega) = \mathcal{C}$, e.g., $\Omega = \frac{\lambda}{2}\|\cdot\|_2^2 + \mathbb{I}_\mathcal{C}$, where $\mathbb{I}_\mathcal{C}(\boldsymbol{\mu}) = +\infty$ if $\boldsymbol{\mu} \notin \mathcal{C}$ and 0 otherwise.

When applying our method, we need to (i) compute the gradient of the convolutional Fenchel–Young loss $L_{\Omega_\tau}(\boldsymbol{\theta}, y)$ with respect to $\boldsymbol{\theta}$ and (ii) draw a sample $\hat{y} \in \hat{\mathcal{Y}}$ from the decoding distribution $\boldsymbol{\pi}(\boldsymbol{\theta}) \in \arg\min_{\boldsymbol{\pi} \in \triangle^N} \Omega^*(\boldsymbol{\theta} + \mathcal{L}^\rho\boldsymbol{\pi})$ to convert $\boldsymbol{\theta} \in \mathbb{R}^d$ into a prediction $\hat{y} \in \hat{\mathcal{Y}}$. Both can be addressed through the following problem, as discussed in Cao et al. (2025, Section 5):

$$\min_{\boldsymbol{\nu} \in \mathcal{V}} \Omega^*(\boldsymbol{\theta} + \boldsymbol{\nu}) \qquad \text{where } \mathcal{V} = \operatorname{conv}\Big(\big\{\boldsymbol{\ell}^\rho(\hat{y}) : \hat{y} \in \hat{\mathcal{Y}}\big\}\Big). \tag{9}$$

By Danskin's theorem (Danskin, 1966), we can compute the gradient of $\Omega^*$ at any $\boldsymbol{\xi} \in \mathbb{R}^d$ as

$$\nabla\Omega^*(\boldsymbol{\xi}) = \arg\max_{\boldsymbol{\mu} \in \mathbb{R}^d}\langle \boldsymbol{\xi}, \boldsymbol{\mu} \rangle - \Omega(\boldsymbol{\mu}).$$

Thus, we can solve this problem using first-order methods.

In the ranking problem in Appendix A.3, $\hat{\mathcal{Y}}$ is the set of permutations of $[d]$, and the loss vector of the NDCG loss is given by $\boldsymbol{\ell}^\rho(\hat{y}) = \big(w_{\hat{y}(1)}, \ldots, w_{\hat{y}(d)}\big)^\top$, where $w_1, \ldots, w_d \geq 0$ are the weights of $d$ documents. Let $\boldsymbol{P}_{\hat{y}} \in \{0,1\}^{d \times d}$ denote the permutation matrix corresponding to $\hat{y} \in \hat{\mathcal{Y}}$. Then, we have $\boldsymbol{\ell}^\rho(\hat{y}) = \boldsymbol{P}_{\hat{y}}\boldsymbol{w}$, where $\boldsymbol{w} = (w_1, \ldots, w_d)^\top$. Consequently, problem (9) can be rewritten as

$$\min_{\boldsymbol{P} \in \mathcal{B}} \Omega^*(\boldsymbol{\theta} + \boldsymbol{P}\boldsymbol{w}) \qquad \text{where } \mathcal{B} = \Big\{\boldsymbol{P} \in \mathbb{R}_{\geq 0}^{d \times d} : \boldsymbol{P}\mathbf{1} = \mathbf{1}, \ \boldsymbol{P}^\top\mathbf{1} = \mathbf{1}\Big\}. \tag{10}$$

The set $\mathcal{B}$ is the so-called Birkhoff polytope, which is the convex hull of the permutation matrices. Consider solving problem (10) with a Frank–Wolfe-type algorithm (e.g., Lacoste-Julien and Jaggi 2015; Garber and Wolf 2021). Using techniques for implementing Carathéodory's theorem (Beck and Shtern 2017; Besançon et al. 2025), we can obtain a solution to problem (10) as a convex combination of $d^2 + 1$ vertices, each of which represents a permutation $\hat{y}$. Let $\boldsymbol{P}^* \in \mathcal{B}$ be an optimal solution to problem (10), $\hat{\mathcal{Y}}^* \subseteq \hat{\mathcal{Y}}$ the set of permutations with non-zero coefficients in the convex combination, and $\{\pi_{\hat{y}}^*\}_{\hat{y} \in \hat{\mathcal{Y}}^*}$ the non-zero coefficients, i.e., $\boldsymbol{P}^* = \sum_{\hat{y} \in \hat{\mathcal{Y}}^*} \pi_{\hat{y}}^* \boldsymbol{P}_{\hat{y}}$. Then, the decoding distribution $\boldsymbol{\pi}(\boldsymbol{\theta})$ is obtained by

setting $\pi_{\hat{y}}(\boldsymbol{\theta})$ to $\pi_{\hat{y}}^*$ for $\hat{y} \in \hat{\mathcal{Y}}^*$ and 0 otherwise. By the envelope theorem (Cao et al., 2025, Lemma 12), the gradient required in (i) can be written as $\nabla \Omega^*(\boldsymbol{\theta} + \mathcal{L}^\rho \boldsymbol{\pi}(\boldsymbol{\theta})) - \boldsymbol{\rho}(y)$, which we can compute as $\nabla \Omega^*(\boldsymbol{\theta} + \boldsymbol{P}^* \boldsymbol{w}) - \boldsymbol{\rho}(y)$, where $\nabla \Omega^*(\boldsymbol{\theta} + \boldsymbol{P}^* \boldsymbol{w}) = \arg \max_{\boldsymbol{\mu} \in \mathbb{R}^d} \langle \boldsymbol{\theta} + \boldsymbol{P}^* \boldsymbol{w}, \boldsymbol{\mu} \rangle - \Omega(\boldsymbol{\mu})$ by Danskin's theorem. Regarding (ii), we can efficiently sample from $\boldsymbol{\pi}(\boldsymbol{\theta})$ by drawing $\hat{y} \in \hat{\mathcal{Y}}^*$ following the distribution $\boldsymbol{\pi}^*$ on $\hat{\mathcal{Y}}^*$.

While we have discussed the case of the ranking problem for concreteness, a similar approach works when a polyhedral representation of $\mathcal{V}$, like the Birkhoff polytope, is available.

## C. Discussion on the Hinge Loss

In Van der Hoeven (2020) and Van der Hoeven et al. (2021), the hinge loss for multiclass classification is introduced as a surrogate loss satisfying the self-bounding property (Assumption 3.3). However, we show that their hinge loss is discontinuous, hence non-convex, and that the standard hinge loss (Crammer and Singer, 2001) is not self-bounding. Therefore, the hinge loss is not a valid surrogate loss for exploiting the surrogate gap.

Consider online classification with $K = d$ classes. For each round $t$, the hinge loss used in Van der Hoeven (2020) and Van der Hoeven et al. (2021) is defined with parameter $\kappa \in [0, 1]$ as follows:

$$L_t(\boldsymbol{W}) = L(\boldsymbol{W}\boldsymbol{x}_t, y_t) = \begin{cases} \max\{1 - m_t(\boldsymbol{W}, y_t), 0\} & \text{if } y_t^* \neq y_t \text{ or } m_t^*(\boldsymbol{W}) \leq \kappa, \\ 0 & \text{if } y_t^* = y_t \text{ and } m_t^*(\boldsymbol{W}) > \kappa, \end{cases}$$

where

$$m_t(\boldsymbol{W}, y) = \langle \mathbf{e}^y, \boldsymbol{W}\boldsymbol{x}_t \rangle - \max_{y' \neq y} \langle \mathbf{e}^{y'}, \boldsymbol{W}\boldsymbol{x}_t \rangle,$$
$$m_t^*(\boldsymbol{W}) = \max_{y \in \mathcal{Y}} m_t(\boldsymbol{W}, y), \qquad \text{and}$$
$$y_t^* \in \arg \max_{y \in \mathcal{Y}} \langle \mathbf{e}^y, \boldsymbol{W}\boldsymbol{x}_t \rangle.$$

Let $\tilde{y}_t \in \arg \max_{y \neq y_t} \langle \mathbf{e}^y, \boldsymbol{W}\boldsymbol{x}_t \rangle$. A subgradient of $L_t$ at $\boldsymbol{W}$ is given by

$$\boldsymbol{G}_t(\boldsymbol{W}) = \begin{cases} (\mathbf{e}^{\tilde{y}_t} - \mathbf{e}^{y_t})\boldsymbol{x}_t^\top & \text{if } y_t^* \neq y_t \text{ or } m_t^*(\boldsymbol{W}) \leq \kappa, \\ 0 & \text{if } y_t^* = y_t \text{ and } m_t^*(\boldsymbol{W}) > \kappa, \end{cases}$$

where 0 means the all-zero matrix.

Below, we argue that the hinge loss with $\kappa < 1$ is discontinuous (though self-bounding) and that the hinge loss with $\kappa = 1$ is not self-bounding (though convex).

**The case of $\kappa < 1$.** Note that we have $m_t(\boldsymbol{W}, y_t) \leq m_t^*(\boldsymbol{W})$. Thus, when $\kappa < 1$, $L_t(\boldsymbol{W})$ viewed as a univariate function of $m_t(\boldsymbol{W}, y_t)$ is discontinuous, where $L_t(\boldsymbol{W}) = 1 - m_t(\boldsymbol{W}, y_t) \geq 1 - \kappa$ for $m_t(\boldsymbol{W}, y_t) \leq \kappa$ and $L_t(\boldsymbol{W}) = 0$ for $m_t(\boldsymbol{W}, y_t) > \kappa$. This implies that $L_t(\boldsymbol{W})$ viewed as a function of $\boldsymbol{W} \in \mathcal{W}$ is discontinuous at the region where $m_t(\boldsymbol{W}, y_t) = \kappa$ holds. The parameter $\kappa$ is set to $1/K$ in Van der Hoeven (2020) and to $1/2$ in Van der Hoeven et al. (2021); therefore, the hinge loss used in their work is discontinuous, hence non-convex.

**The case of $\kappa = 1$.** The choice of $\kappa = 1$ corresponds to the standard convex hinge loss for multiclass classification. However, this is not self-bounding. Suppose $\|\boldsymbol{x}_t\|_2 = 1$. Then, if $y_t^* = y_t$ and $m_t^*(\boldsymbol{W}) \leq \kappa = 1$, we have $\|\boldsymbol{G}_t(\boldsymbol{W})\|_F^2 = 2\|\boldsymbol{x}_t\|_2^2 = 2$. On the other hand, the loss value, $L_t(\boldsymbol{W}) = 1 - m_t^*(\boldsymbol{W})$, can be arbitrarily close to zero as $m_t^*(\boldsymbol{W})$ approaches $\kappa = 1$, preventing us from ensuring $\|\boldsymbol{G}_t(\boldsymbol{W})\|_F^2 \lesssim L_t(\boldsymbol{W})$. Therefore, the hinge loss with $\kappa = 1$ is not self-bounding.

## D. Proof of Proposition 2.4

*Proof.* By the standard analysis of OGD (e.g., Orabona 2025, Theorem 2.13), for any $t$ and $\boldsymbol{U}_t \in \mathcal{W}$, we have

$$L_t(\boldsymbol{W}_t) - L_t(\boldsymbol{U}_t) \leq \frac{\|\boldsymbol{W}_t - \boldsymbol{U}_t\|_F^2 - \|\boldsymbol{W}_{t+1} - \boldsymbol{U}_t\|_F^2}{2\eta_t} + \frac{\eta_t}{2}\|\boldsymbol{G}_t\|_F^2.$$

Summing over $t$, we obtain

$$\sum_{t=1}^{T}(L_t(\boldsymbol{W}_t) - L_t(\boldsymbol{U}_t)) \leq \underbrace{\sum_{t=1}^{T}\frac{\|\boldsymbol{W}_t - \boldsymbol{U}_t\|_{\mathrm{F}}^2 - \|\boldsymbol{W}_{t+1} - \boldsymbol{U}_t\|_{\mathrm{F}}^2}{2\eta_t}}_{(A)} + \sum_{t=1}^{T}\frac{\eta_t}{2}\|\boldsymbol{G}_t\|_{\mathrm{F}}^2.$$

Below, we define $\boldsymbol{U}_{T+1} = \boldsymbol{U}_T$ for convenience. For $t = 1, \ldots, T$, it holds that

$$\begin{aligned}
\|\boldsymbol{W}_{t+1} - \boldsymbol{U}_t\|_{\mathrm{F}}^2 &= \|\boldsymbol{W}_{t+1} - \boldsymbol{U}_{t+1}\|_{\mathrm{F}}^2 + \|\boldsymbol{U}_{t+1} - \boldsymbol{U}_t\|_{\mathrm{F}}^2 + 2\langle\boldsymbol{W}_{t+1} - \boldsymbol{U}_{t+1}, \boldsymbol{U}_{t+1} - \boldsymbol{U}_t\rangle \\
&\geq \|\boldsymbol{W}_{t+1} - \boldsymbol{U}_{t+1}\|_{\mathrm{F}}^2 + 2\langle\boldsymbol{W}_{t+1} - \boldsymbol{U}_{t+1}, \boldsymbol{U}_{t+1} - \boldsymbol{U}_t\rangle && \text{Ignoring } \|\boldsymbol{U}_{t+1} - \boldsymbol{U}_t\|_{\mathrm{F}}^2 \geq 0 \\
&\geq \|\boldsymbol{W}_{t+1} - \boldsymbol{U}_{t+1}\|_{\mathrm{F}}^2 - 2D\|\boldsymbol{U}_{t+1} - \boldsymbol{U}_t\|_{\mathrm{F}}. && \text{Cauchy–Schwarz and } \|\boldsymbol{W}_{t+1} - \boldsymbol{U}_{t+1}\|_{\mathrm{F}} \leq D
\end{aligned}$$

Therefore, we have

$$(A) \leq \underbrace{\sum_{t=1}^{T}\frac{\|\boldsymbol{W}_t - \boldsymbol{U}_t\|_{\mathrm{F}}^2 - \|\boldsymbol{W}_{t+1} - \boldsymbol{U}_{t+1}\|_{\mathrm{F}}^2}{2\eta_t}}_{(B)} + \underbrace{\sum_{t=1}^{T}\frac{D}{\eta_t}\|\boldsymbol{U}_{t+1} - \boldsymbol{U}_t\|_{\mathrm{F}}}_{(C)}.$$

Since $\eta_t$ is non-increasing, $\eta_t \geq \eta_T$ holds, and thus (C) is at most $\frac{D}{\eta_T}P_T$. We bound (B) from above as follows:

$$\begin{aligned}
(B) &= \frac{\|\boldsymbol{W}_1 - \boldsymbol{U}_1\|_{\mathrm{F}}^2}{2\eta_1} + \frac{1}{2}\sum_{t=2}^{T}\left(\frac{1}{\eta_t} - \frac{1}{\eta_{t-1}}\right)\|\boldsymbol{W}_t - \boldsymbol{U}_t\|_{\mathrm{F}}^2 - \frac{\|\boldsymbol{W}_{T+1} - \boldsymbol{U}_{T+1}\|_{\mathrm{F}}^2}{2\eta_T} && \text{Summation by parts} \\
&\leq \frac{\|\boldsymbol{W}_1 - \boldsymbol{U}_1\|_{\mathrm{F}}^2}{2\eta_1} + \frac{1}{2}\sum_{t=2}^{T}\left(\frac{1}{\eta_t} - \frac{1}{\eta_{t-1}}\right)\|\boldsymbol{W}_t - \boldsymbol{U}_t\|_{\mathrm{F}}^2 && \text{Ignoring the last term} \\
&\leq \frac{D^2}{2\eta_1} + \frac{D^2}{2}\sum_{t=2}^{T}\left(\frac{1}{\eta_t} - \frac{1}{\eta_{t-1}}\right) && \eta_{t-1} \geq \eta_t \text{ and } \|\boldsymbol{W}_t - \boldsymbol{U}_t\|_{\mathrm{F}} \leq D \\
&= \frac{D^2}{2\eta_1} + \frac{D^2}{2}\left(\frac{1}{\eta_T} - \frac{1}{\eta_1}\right) && \text{Telescoping} \\
&= \frac{D^2}{2\eta_T}.
\end{aligned}$$

Therefore, we can bound (A) from above by $\frac{D}{\eta_T}\left(\frac{D}{2} + P_T\right)$, obtaining the desired bound. $\qquad\square$

## E. Numerical Experiments

We experimentally evaluate the performance of our proposed Polyak-style learning rate (see Section 3.4) in comparison to other learning rates. Experiments were conducted on Google Colab with an AMD EPYC 7B12 CPU, 12 GB RAM, running Ubuntu 22.04.

**Setup.** We consider online $K$-class classification over $T = 10,000$ rounds with the logistic surrogate loss:

$$L(\boldsymbol{W}; \boldsymbol{x}_t, y_t) = -\log_2\left(\frac{\exp(\langle\mathbf{e}_{y_t}, \boldsymbol{W}\boldsymbol{x}_t\rangle)}{\sum_{y\in[K]}\exp(\langle\mathbf{e}_y, \boldsymbol{W}\boldsymbol{x}_t\rangle)}\right),$$

where $\boldsymbol{W} \in \mathcal{W} \subseteq \mathbb{R}^{K\times m}$ is a linear estimator, $y_t \in \{1, \ldots, K\}$ indicates a label, and $\boldsymbol{x}_t \in \mathbb{R}^m$ is an input feature vector. This loss function satisfies the self-bounding property in Assumption 3.3 with $M = \frac{1}{\ln 2}$ (Van der Hoeven, 2020, Lemma 2). As the evaluation metric, we report the cumulative 0-1 loss (number of mistakes). Each configuration given below is repeated for 10 independent trials, and we plot the mean and the 95% confidence intervals.

**Datasets.** We consider two types of datasets: synthetic datasets and MNIST datasets (LeCun et al., 1998).

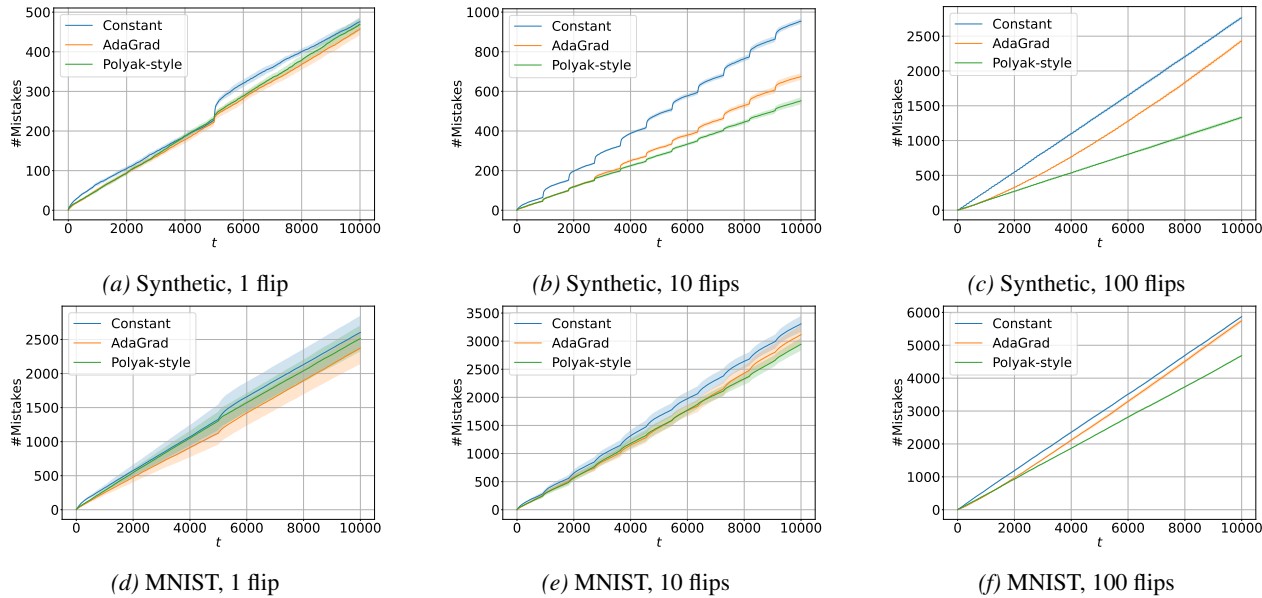

*Figure 2.* Experimental results on synthetic and MNIST datasets for different learning rates under varying numbers of label flips.

- **Synthetic datasets:** We set $K = 2$ and $m = 2$, and use a label set of $\{-1, +1\}$, rather than $\{1, 2\}$. Accordingly, we represent the linear estimator as a vector $\boldsymbol{w} \in \mathcal{W} \subseteq \mathbb{R}^2$. Each feature vector $\boldsymbol{x}_t \in \mathbb{R}^m$ is an input feature vector drawn uniformly from the two-dimensional unit sphere. We define a reference vector $\boldsymbol{u} = \frac{1}{\sqrt{2}}(1, 1)^\top$. In the stationary environment, labels are given by $y_t = +1$ if $\langle \boldsymbol{u}, \boldsymbol{x}_t \rangle \geq 0$ and $-1$ otherwise. Non-stationarity is simulated by introducing segment-wise label flips: the number of flips is set to $\{1, 10, 100\}$, and at each flip all labels are multiplied by $-1$. Consequently, the true estimator in each segment alternates between $\boldsymbol{u}$ and $-\boldsymbol{u}$.

- **MNIST datasets:** We set $K = 3$ and $m = 784$ ($28 \times 28$ images flattened into vectors). Each feature vector $\boldsymbol{x}_t \in \mathbb{R}^m$ is obtained by normalizing the pixel values of an image to have a unit $\ell_2$-norm. Similar to the synthetic datasets, we simulate non-stationarity by introducing segment-wise label flips with counts of $\{1, 10, 100\}$. For each segment, we uniformly sample three distinct digits. These three digits constitute the set of labels in that segment. Within each segment, we sample images of the three digits uniformly with replacement and use them as the input feature vectors $\boldsymbol{x}_t$.

**Methods.** We run OGD with projection onto $\mathcal{W}$, which is an $\ell_2$-ball with a diameter of $D = 20$. Although the true estimator may lie in the unit ball, its exact scale is typically unknown a priori. We thus adopt a larger diameter of $D = 20$ to reflect this uncertainty. The decoding method used in this section is that of Van der Hoeven et al. (2021). This decoding approach, when applied to $K$-class classification, yields a surrogate gap of $\alpha = 1/K$. We compare the following learning rates of OGD:

- **Constant**: a fixed value of $\eta_t = \frac{\alpha}{M}$, which satisfies (4) in Theorem 3.4. This approach was adopted by Van der Hoeven (2020) and Sakaue et al. (2024).

- **AdaGrad**: a standard adaptive learning rate of $\eta_t = \frac{D}{\sqrt{2 \sum_{s=1}^{t} \|\nabla L(\boldsymbol{w}_s; \boldsymbol{x}_s, y_s)\|_2^2}}$ (see e.g., Orabona 2025, Theorem 4.29). This does not necessarily satisfy (4) in Theorem 3.4.

- **Polyak-style**: our proposed learning rate inspired by Polyak's rule, which satisfies (4) in Theorem 3.4.

**Results.** Figure 2 shows the results for synthetic datasets and MNIST datasets. The curves show the cumulative number of mistakes with 95% confidence intervals over 10 independent trials. For the "1 flip" setting (nearly stationary), "Polyak-style" performs slightly worse than AdaGrad, yet remains competitive. For "10 flips" (moderately non-stationary), the constant learning rate performs worst, AdaGrad is intermediate, and Polyak-style outperforms them. For "100 flips" (highly non-stationary), the performance gap between Polyak-style and the others becomes more pronounced. In summary, the Polyak-style learning rate consistently achieves competitive or superior performance, with its advantage becoming more evident as the level of non-stationarity increases.

# F. Missing Proofs and Discussion in Section 4

### F.1. Proof of Lemma 4.3

*Proof.* As shown in Cao et al. (2025, Lemma 8), the conjugacy of addition and infimal convolution implies $\Omega_\tau^*(\boldsymbol{\theta}) = (\Omega + \tau)^*(\boldsymbol{\theta}) = \inf\{\Omega^*(\boldsymbol{\theta} - \boldsymbol{\theta}') + \tau^*(\boldsymbol{\theta}') : \boldsymbol{\theta}' \in \mathbb{R}^d\} = \Omega^*(\boldsymbol{\theta} + \mathcal{L}^\rho \boldsymbol{\pi}(\boldsymbol{\theta}))$. Thus, we have

$$
\begin{aligned}
L_{\Omega_\tau}(\boldsymbol{\theta}, y) &= \Omega^*(\boldsymbol{\theta} + \mathcal{L}^\rho \boldsymbol{\pi}(\boldsymbol{\theta})) + \Omega(\boldsymbol{\rho}(y)) + \tau(\boldsymbol{\rho}(y)) - \langle \boldsymbol{\theta}, \boldsymbol{\rho}(y) \rangle \\
&= \Omega^*(\boldsymbol{\theta} + \mathcal{L}^\rho \boldsymbol{\pi}(\boldsymbol{\theta})) + \Omega(\boldsymbol{\rho}(y)) - \langle \boldsymbol{\theta} + \mathcal{L}^\rho \boldsymbol{\pi}(\boldsymbol{\theta}), \boldsymbol{\rho}(y) \rangle + \langle \mathcal{L}^\rho \boldsymbol{\pi}(\boldsymbol{\theta}), \boldsymbol{\rho}(y) \rangle + \tau(\boldsymbol{\rho}(y)) \\
&= L_\Omega(\boldsymbol{\theta} + \mathcal{L}^\rho \boldsymbol{\pi}(\boldsymbol{\theta}), y) + \langle \mathcal{L}^\rho \boldsymbol{\pi}(\boldsymbol{\theta}), \boldsymbol{\rho}(y) \rangle + \tau(\boldsymbol{\rho}(y)),
\end{aligned}
$$

where the last equality comes from the definition of the Fenchel–Young loss with the regularizer $\Omega$. Therefore, it remains to show $\langle \mathcal{L}^\rho \boldsymbol{\pi}(\boldsymbol{\theta}), \boldsymbol{\rho}(y) \rangle + \tau(\boldsymbol{\rho}(y)) = \mathbb{E}_{\hat{y} \sim \boldsymbol{\pi}(\boldsymbol{\theta})}[\ell(\hat{y}, y)]$. From $\min_{\hat{y} \in \hat{\mathcal{Y}}} \ell(\hat{y}, y) = 0$ in Assumption 2.2 and $-\min_{\hat{y} \in \hat{\mathcal{Y}}} \langle \boldsymbol{\rho}(y), \boldsymbol{\ell}^\rho(\hat{y}) \rangle = \tau(\boldsymbol{\rho}(y))$ in Definition 4.1, we have

$$
\min_{\hat{y} \in \hat{\mathcal{Y}}} \ell(\hat{y}, y) = \min_{\hat{y} \in \hat{\mathcal{Y}}} \langle \boldsymbol{\rho}(y), \boldsymbol{\ell}^\rho(\hat{y}) \rangle + c(y) = -\tau(\boldsymbol{\rho}(y)) + c(y) = 0,
$$

hence $\tau(\boldsymbol{\rho}(y)) = c(y)$. Therefore, we obtain

$$
\langle \mathcal{L}^\rho \boldsymbol{\pi}(\boldsymbol{\theta}), \boldsymbol{\rho}(y) \rangle + \tau(\boldsymbol{\rho}(y)) = \langle \mathcal{L}^\rho \boldsymbol{\pi}(\boldsymbol{\theta}), \boldsymbol{\rho}(y) \rangle + c(y) = \sum_{\hat{y} \in \hat{\mathcal{Y}}} \pi_{\hat{y}}(\boldsymbol{\theta})(\langle \boldsymbol{\rho}(y), \boldsymbol{\ell}^\rho(\hat{y}) \rangle + c(y)) = \mathbb{E}_{\hat{y} \sim \boldsymbol{\pi}(\boldsymbol{\theta})}[\ell(\hat{y}, y)],
$$

as desired. $\square$

### F.2. Discussion on the Connection to Cao et al. (2025)

We discuss the connection between Lemma 4.3 and the excess risk analysis of Cao et al. (2025). For a distribution $\boldsymbol{\eta} \in \triangle^K$, write $\bar{\boldsymbol{\rho}}_{\boldsymbol{\eta}} = \mathbb{E}_{y \sim \boldsymbol{\eta}}[\boldsymbol{\rho}(y)]$. We also use the Fenchel–Young-loss expression

$$
L_\Omega(\boldsymbol{\xi}, \boldsymbol{\mu}) = \Omega^*(\boldsymbol{\xi}) + \Omega(\boldsymbol{\mu}) - \langle \boldsymbol{\xi}, \boldsymbol{\mu} \rangle \qquad (\boldsymbol{\mu} \in \mathcal{C}),
$$

which extends the notation $L_\Omega(\boldsymbol{\xi}, y)$ by replacing $\boldsymbol{\rho}(y)$ with $\boldsymbol{\mu}$. In the proof of Cao et al. (2025, Lemma 14), for any $\boldsymbol{\eta} \in \triangle^K$ and $\boldsymbol{\theta} \in \mathbb{R}^d$, they show

$$
\mathbb{E}_{y \sim \boldsymbol{\eta}}[L_{\Omega_\tau}(\boldsymbol{\theta}, y)] - \inf_{\boldsymbol{\theta}' \in \mathbb{R}^d} \mathbb{E}_{y \sim \boldsymbol{\eta}}[L_{\Omega_\tau}(\boldsymbol{\theta}', y)] = L_\Omega(\boldsymbol{\theta} + \mathcal{L}^\rho \boldsymbol{\pi}(\boldsymbol{\theta}), \bar{\boldsymbol{\rho}}_{\boldsymbol{\eta}}) + \mathbb{E}_{\hat{y} \sim \boldsymbol{\pi}(\boldsymbol{\theta})}\left[\mathbb{E}_{y \sim \boldsymbol{\eta}}[\ell(\hat{y}, y)]\right] - \min_{\hat{y} \in \hat{\mathcal{Y}}} \mathbb{E}_{y \sim \boldsymbol{\eta}}[\ell(\hat{y}, y)].
$$

The proof of this equality relies on $\Omega_\tau^*(\boldsymbol{\theta}) = \Omega^*(\boldsymbol{\theta} + \mathcal{L}^\rho \boldsymbol{\pi}(\boldsymbol{\theta}))$ (Cao et al., 2025, Lemma 8), which is also the key identity in our proof of Lemma 4.3. By dropping the nonnegative term $L_\Omega(\boldsymbol{\theta} + \mathcal{L}^\rho \boldsymbol{\pi}(\boldsymbol{\theta}), \bar{\boldsymbol{\rho}}_{\boldsymbol{\eta}}) \geq 0$, the authors obtained

$$
\mathbb{E}_{\hat{y} \sim \boldsymbol{\pi}(\boldsymbol{\theta})}\left[\mathbb{E}_{y \sim \boldsymbol{\eta}}[\ell(\hat{y}, y)]\right] - \min_{\hat{y} \in \hat{\mathcal{Y}}} \mathbb{E}_{y \sim \boldsymbol{\eta}}[\ell(\hat{y}, y)] \leq \mathbb{E}_{y \sim \boldsymbol{\eta}}[L_{\Omega_\tau}(\boldsymbol{\theta}, y)] - \inf_{\boldsymbol{\theta}' \in \mathbb{R}^d} \mathbb{E}_{y \sim \boldsymbol{\eta}}[L_{\Omega_\tau}(\boldsymbol{\theta}', y)],
$$

that is, a linear upper bound on the target excess risk in terms of the surrogate excess risk. At this point, the proof of our Lemma 4.3 deviates from their original proof. First, instead of dropping this nonnegative term, we retain its specialization at a ground-truth label, which plays a crucial role in our analysis through Lemma 4.4. Second, we are interested in evaluating the target and surrogate losses on a ground-truth label, rather than on a distribution $\boldsymbol{\eta}$. Therefore, we can simplify the expression by focusing on the case of $\boldsymbol{\eta} = \mathbf{e}^y$ for ground-truth $y \in \mathcal{Y}$, where $\bar{\boldsymbol{\rho}}_{\boldsymbol{\eta}} = \boldsymbol{\rho}(y)$. This implies $\inf_{\boldsymbol{\theta}' \in \mathbb{R}^d} \mathbb{E}_{y \sim \boldsymbol{\eta}}[L_{\Omega_\tau}(\boldsymbol{\theta}', y)] = 0$ and $\min_{\hat{y} \in \hat{\mathcal{Y}}} \mathbb{E}_{y \sim \boldsymbol{\eta}}[\ell(\hat{y}, y)] = 0$ under Assumption 2.2. Consequently, we obtain the target–surrogate relation in Lemma 4.3:

$$
L_{\Omega_\tau}(\boldsymbol{\theta}, y) = L_\Omega(\boldsymbol{\theta} + \mathcal{L}^\rho \boldsymbol{\pi}(\boldsymbol{\theta}), y) + \mathbb{E}_{\hat{y} \sim \boldsymbol{\pi}(\boldsymbol{\theta})}[\ell(\hat{y}, y)].
$$

Thus, while our Lemma 4.3 is built on an analysis similar to that of Cao et al. (2025), it is tailored for our purpose of deriving the "small-surrogate-loss + path-length" bound.

### F.3. Proof of Lemma 4.4

*Proof.* From $\text{dom}(\Omega) = \mathcal{C}$ and Danskin's theorem (Danskin, 1966), we have $\nabla\Omega^*(\boldsymbol{\theta} + \mathcal{L}^\rho\boldsymbol{\pi}(\boldsymbol{\theta})) = \arg\max_{\boldsymbol{\mu}\in\mathcal{C}}\{\langle\boldsymbol{\theta} + \mathcal{L}^\rho\boldsymbol{\pi}(\boldsymbol{\theta}), \boldsymbol{\mu}\rangle - \Omega(\boldsymbol{\mu})\}$.[9] Since $\Omega$ is $\lambda$-strongly convex over $\mathcal{C} = \text{conv}(\{\boldsymbol{\rho}(y) \in \mathbb{R}^d : y \in \mathcal{Y}\})$, the quadratic lower bound on the (standard) Fenchel–Young loss (Blondel et al., 2022, Proposition 3) implies

$$L_\Omega(\boldsymbol{\theta} + \mathcal{L}^\rho\boldsymbol{\pi}(\boldsymbol{\theta}), y) \geq \frac{\lambda}{2}\|\nabla\Omega^*(\boldsymbol{\theta} + \mathcal{L}^\rho\boldsymbol{\pi}(\boldsymbol{\theta})) - \boldsymbol{\rho}(y)\|_2^2.$$

Furthermore, the envelope theorem (Cao et al., 2025, Lemma 12) implies

$$\nabla L_{\Omega_\tau}(\boldsymbol{\theta}, y) = \nabla\Omega^*(\boldsymbol{\theta} + \mathcal{L}^\rho\boldsymbol{\pi}(\boldsymbol{\theta})) - \boldsymbol{\rho}(y).$$

Substituting this into the above quadratic lower bound yields the desired inequality. $\square$

## G. Proof of the Lower Bound

*Proof of Theorem 5.1.* Fix nonnegative integers $T_\mathsf{F}, T_\mathsf{P}$ with $T = T_\mathsf{F} + T_\mathsf{P} > 0$. Let $K = d = 2$, $\mathcal{Y} = \hat{\mathcal{Y}} = \{-1, +1\}$, and $\boldsymbol{x}_t = (1, 0)^\top$ for $t = 1, \ldots, T$. Set each ground-truth label $y_t$ to $-1$ or $+1$ independently and uniformly. Then, any deterministic learner inevitably incurs $\mathbb{E}[Z_T] = T/2$, where $Z_T = \sum_{t=1}^T \ell(\hat{y}_t, y_t) = \sum_{t=1}^T \mathbb{1}_{\hat{y}_t \neq y_t}$. By Yao's principle (Yao, 1977), for any possibly randomized learner, there exists a worst-case instance such that $\mathbb{E}[Z_T] \geq T/2$, where the expectation is taken over the learner's possible randomness. Below, we let $y_1, \ldots, y_T$ be the labels of such an instance.

For the above instance, we design comparator sequences to show the claim. Let $\mathcal{W} = \{\boldsymbol{W} \in \mathbb{R}^{2\times 2} : \|\boldsymbol{W}\|_\mathsf{F} \leq 1\}$ be the domain and $\boldsymbol{U}_1, \ldots, \boldsymbol{U}_T \in \mathcal{W}$ denote comparators. For convenience, let $\zeta_t = y_t\langle(+1, -1)^\top, \boldsymbol{U}_t\boldsymbol{x}_t\rangle$. Then, the smooth hinge loss (see Appendix B.1) satisfies

$$L_t(\boldsymbol{U}_t) = \begin{cases} 1 - 2\zeta_t & \text{if } \zeta_t \leq 0, \\ (1 - \zeta_t)^2 & \text{if } 0 < \zeta_t < 1, \\ 0 & \text{if } 1 \leq \zeta_t. \end{cases}$$

We use two elementary comparators,

$$\boldsymbol{U}^+ = \frac{1}{2}\begin{bmatrix} +1 & +1 \\ -1 & +1 \end{bmatrix} \quad \text{and} \quad \boldsymbol{U}(y) = \frac{1}{2}\begin{bmatrix} +y & +1 \\ -y & +1 \end{bmatrix} \quad (y \in \{-1, +1\}).$$

The comparator sequence is defined using the fixed comparator $\boldsymbol{U}^+$ in the first $T_\mathsf{F}$ rounds and the label-fitting comparator $\boldsymbol{U}(y_t)$ in the remaining $T_\mathsf{P}$ rounds:

$$\boldsymbol{U}_t = \begin{cases} \boldsymbol{U}^+ & \text{if } t \leq T_\mathsf{F}, \\ \boldsymbol{U}(y_t) & \text{if } t > T_\mathsf{F}. \end{cases}$$

For $t \leq T_\mathsf{F}$, we have $\zeta_t = y_t$, and hence $L_t(\boldsymbol{U}_t)$ is either 0 or 3. For $t > T_\mathsf{F}$, we have $\zeta_t = 1$, and hence $L_t(\boldsymbol{U}_t) = 0$. Therefore, we have

$$F_T = \sum_{t=1}^T L_t(\boldsymbol{U}_t) \leq 3T_\mathsf{F}.$$

As for the path length, the comparator is constant in the first $T_\mathsf{F}$ rounds, and every subsequent change has Frobenius norm at most $\sqrt{2}$. There are at most $T_\mathsf{P}$ such changes, including the possible transition from the fixed-comparator block to the label-fitting block. Thus, it holds that

$$P_T = \sum_{t=2}^T \|\boldsymbol{U}_t - \boldsymbol{U}_{t-1}\|_\mathsf{F} \leq \sqrt{2}T_\mathsf{P}.$$

Combining these bounds with $\mathbb{E}[Z_T] \geq T/2$, we obtain

$$\mathbb{E}[Z_T] \geq \frac{T_\mathsf{F} + T_\mathsf{P}}{2} \geq \frac{F_T + P_T}{6},$$

where the last inequality uses $F_T + P_T \leq 3T_\mathsf{F} + \sqrt{2}T_\mathsf{P} \leq 3(T_\mathsf{F} + T_\mathsf{P})$. This proves the claimed $\Omega(F_T + P_T)$ lower bound on the expected cumulative target loss. $\square$

---

[9] We assumed $\text{dom}(\Omega) = \mathcal{C}$ in Assumption 4.2 solely for ease of establishing this relation.

