# OpenReview forum: "Non-Stationary Online Structured Prediction with Surrogate Losses"
_ICML.cc/2026/Conference — ICML 2026 regular_

### Official Review · Reviewer_3s6r · 2026-03-02

**Soundness:** 4
**Presentation:** 4
**Significance:** 3
**Originality:** 3
**Overall Recommendation:** 5
**Confidence:** 3

**Summary:**

Summary: This paper studies online structured prediction under non-stationarity using surrogate loss framework (learn a score vector via continuous convex optimization, then decode to a discrete prediction). Prior work studies the stationary environment against the best fixed predictor, which can perform badly in non-stationary environments (as bad as $\Omega(T)$. The main goal of this paper is to bound the cumulative target loss by quantities that remain small when the environment changes slowly, e.g. piecewise stationary environments. They prove bounds of the form $\sum_{t=1}^T\ell(\hat{y}_t,y_t)\leq F_T+O(1+P_T)$, where $F_T$ is the cumulative loss of that is related to the surrogate loss type and $P_T$ is the path length (a measure of non-stationarity).

Contribution:
1. Small surrogate loss plus path length bound via surrogate gap. They prove that with OGD and a learning rate scheme, the expected cumulative regret is bounded by $F_T+O(1+P_T)$.

2. They propose a Polyak learning rate rule, which is more adaptive.

3. They use convolutional Fenchel-Young loss, to address more general settings where previous assumptions are not satisfied.

4. They provide matching lower bounds.

**Compliance With Llm Reviewing Policy:**

Affirmed.

**Final Justification:**

Based on the authors' responses which address our concerns, we will keep our positive score (5).

**Key Questions For Authors:**

Can the authors demonstrate more on how the finite space $\mathcal{Y}$ is harder than continuous spaces by referencing literature or providing examples?

Are there any theoretical/empirical guarantees for the Polyak learning rate?

Can the authors explain how $F_T$ scales with $T$ by some examples? For example, is it constant or sublinear?

Is it possible to extend the results to more non-stationary settings beyond path length metric?

**Limitations:**

yes

**Strengths And Weaknesses:**

Strength:

This paper is technically sound, meaning it does not have substantial mistakes or flaws from the theoretical side.

This paper is clear and easy to read, with clear main results roadmaps on the formulation, derivations, theoretical guarantees. Although it may be a bit abstract in the main body, it provides extensive examples to convince the readers that the assumptions and definitions make sense.

Significance: This paper address a gap in the literature, as described in its title.

Originality: The small surrogate loss plus path length bound seems original; The design of convolutional Fenchel Young loss with applications to the more general setting is also original.

Weakness:

As authors noted, explorations on bandit feedback is an important direction.

The results relies on strong structural assumptions, e.g. Assumption 3.1-3.3.

Strong convexity parameter $\lambda$: as authors notes, $\lambda$ should  be chosen to balance the tradeoff appeared in the regret bound. Theoretical or empirical guidance may be built on this aspect.

It seems that Section 3.4 does not have theoretical guarantees.

More general nonstationary settings: using path length as a measure of non-stationarity, which can be still linearly in $T$ under some cases. It could be better if the authors can explore more advanced non-stationary metric, e.g. segmentation (Huang and Wang, 2025) to hedge against more adversarial non-stationary environments.

Chengpiao Huang, Kaizheng Wang (2025) A Stability Principle for Learning Under Nonstationarity. Operations Research 73(6):3044-3064.

---

> ### Author Rebuttal · Authors · 2026-03-25
>
> We sincerely thank the reviewer for the thorough and positive review. We address the questions and weaknesses below.
>
> ## **Q1. How is finite $\hat{\mathcal{Y}}$ harder than continuous spaces?**
> We clarify that we do not intend to claim that a finite prediction space is harder than a continuous one; these are inherently different and incomparable. Rather, as stated in our introduction, the discrete nature of $\mathcal{Y}$ (and $\hat{\mathcal{Y}}$) in structured prediction prevents direct learning of mappings via efficient continuous optimization. The standard remedy is the surrogate loss framework (Bartlett et al., 2006; Blondel et al., 2020), which introduces an intermediate continuous score space $\mathbb{R}^d$ and enables us to learn an estimator $W_t\colon\mathcal{X}\to\mathbb{R}^d$ via continuous optimization. Our work builds on this framework.
>
> ## **Q2. Theoretical and empirical guarantees for the Polyak-style learning rate**
> The Polyak-style learning rate in eq. (5) does carry theoretical guarantees. The step-size rule is designed so that $\eta_t$ satisfies the conditions required by our main theorems, eq. (4) in Theorem 3.4 and eq. (8) in Theorem 4.5, thereby ensuring $\sum_t\ell_t\le F_T+O(1+P_T)$.
>
> Empirically, compared to the conservative fixed choice $\eta$ (used by Van der Hoeven 2020 and Sakaue et al. 2024), the Polyak rule can be more aggressive, leading to faster adaptation to non-stationarity. Appendix E confirms a consistent and substantial improvement over fixed and AdaGrad-type step sizes, with the benefit growing as the environment becomes more non-stationary.
>
> ## **Q3. How does $F_T$ scale with $T$?**
> $F_T=\sum_{t=1}^TL_t(U_t)$ is the cumulative surrogate loss of the comparator sequence $\\{U_t\\}\_{t=1}^T$, where the sequence of $U_t$ is *arbitrary*. Its scaling therefore trades off against the path length $P_T=\sum_{t=2}^{T}\\|U_t-U_{t-1}\\|$.
> For standard surrogate losses $L_t$ with a separation margin (e.g., the smoothed hinge), there exists $U_t$ with $L_t(U_t)=0$ for each $t$ (hence $F_T=0$), but the sequence of such $U_t$ may have a large path length $P_T$. Conversely, if we fix $U_t$ to some $U$ for all $t$, then $P_T=0$ but $F_T$ may grow linearly with $T$.
>
> Our target-loss bound of $F_T+O(P_T)$ is particularly effective relative to the prior results (Van der Hoeven 2020; Van der Hoeven et al. 2021; Sakaue et al. 2024), which only consider fixed $U$, in the following typical non-stationary scenarios:
> the $T$ rounds are divided into a small number of, say $J$, segments, each of which has a fixed comparator $U_t=U^{(j)}$ with zero (or small) surrogate losses for all $t$ within the segment (i.e., stationary), but the comparators differ across segments (i.e., jumps), leading to a target loss bound of
> $$F_T+O(P_T)\approx0+O\left(\sum_{j=2}^J\\|U^{(j)}-U^{(j-1)}\\|\right)=O(JD),$$
> where $D$ is the diameter of the estimator domain $\mathcal{W}$. In contrast, the prior results would yield $F_T\simeq T$ by fixing $U_t=U$ for all $t$, resulting in a trivial target loss bound of $O(T)$.
>
> ## **Q4. Extension to non-stationary settings beyond path length**
> We thank the reviewer for pointing us to Huang and Wang (2025). Extending to segment-based or stability-based non-stationarity measures is an interesting and important future direction. Our current analysis relies on path length $P_T=\sum_t\\|U_t-U_{t-1}\\|$ since the dynamic regret decomposition naturally involves comparator shifts $\\|U_t-U_{t-1}\\|$ (cf. Proposition 2.4). Incorporating other non-stationarity measures would require a fundamentally different algorithm and analysis. We will include a discussion of this direction in the revised version, citing the suggested and related references.
>
> **Note on the pessimism of the path variation.**
> As in our answer to Q3, our path-length definition is valid for arbitrary comparator sequences $\\{U_t\\}_t$ (as with that in existing studies on *universal* dynamic regret, e.g., Zhao et al. 2024). This contrasts with the path variation $V_T$ considered in Huang and Wang (2025), which is defined only for the sequence of minimizers. For example, in our response to Q3, we may regard each segment as *quasi*-stationary segments (Huang and Wang, 2025). As such, the path-length $P_T$ we adopt can bypass the pessimism of $V_T$ noted by Huang and Wang (2025): $V_T$ tracks exact minimizers and may grow linearly in $T$ even under small fluctuations within segments, whereas our $P_T$ allows the comparator to be held constant within each segment. Further inspection of how such choices of $U_t$ affect $F_T$ may yield results related to those obtained via segment-based measures, which we leave as an interesting future direction.
>
> ## **On the choice of $\lambda$ (in Weaknesses)**
> We clarify that we can determine $\lambda$, the strong convexity of $\Omega$, by the choice of surrogate loss—e.g., $\lambda=1$ for the logistic loss. Our bounds hold with fixed $\lambda$, while its adaptive control is future work.

---

> > ### Author Rebuttal · Reviewer_3s6r · 2026-03-31
> >
> > We thank the authors for their responses, which address our questions. We will keep our score.

---

### Official Review · Reviewer_zVFu · 2026-03-12

**Soundness:** 3
**Presentation:** 3
**Significance:** 3
**Originality:** 3
**Overall Recommendation:** 4
**Confidence:** 1

**Summary:**

The paper studies non-stationary online structured prediction with surrogate losses. Its main result is a cumulative target-loss bound of the form $F_T + O(1+P_T)$, where $F_T$ is the cumulative surrogate loss of a comparator sequence and $P_T$ is its path length. The analysis combines dynamic-regret bounds for online gradient descent with surrogate-gap arguments, proposes a Polyak-style learning-rate rule, extends the framework to convolutional Fenchel-Young losses for more general structured prediction settings, and includes a lower bound.

**Compliance With Llm Reviewing Policy:**

Affirmed.

**Key Questions For Authors:**

1. Section 4 seems to extend the framework beyond assumptions. Do the authors view this section mainly as a theoretical extension, or do they expect it to be practically useful as well (other than NDCG)?
2. In the experiments, the comparison seems to be mostly between different learning-rate choices for OGD. Could the authors comment on why stronger adaptive baselines for non-stationary online learning were not included?
3. In the broader structured-prediction setting, is the proposed learning-rate rule computationally practical when the output space is large?

**Limitations:**

yes

**Strengths And Weaknesses:**

**Strength**

The paper addresses an interesting problem, namely non-stationary online structured prediction, and presents a clean target bound in terms of comparator surrogate loss and path length. Narrative appears well structured with some clear connection with a clear connection between regret analysis. The manuscript is also generally well organized.

**Weaknesses**
The empirical validation appears limited relative to the scope of the theory. In particular, the experiments seem to focus on multiclass classification, and I did not see empirical validation of the broader structured-prediction extension developed later in the paper. However, I understand that the main contribution is theory.

Given my confidence level, I am not certain that I checked all technical details of the proofs closely enough to rule out hidden issues.

---

> ### Author Rebuttal · Authors · 2026-03-25
>
> We sincerely thank the reviewer for the careful and constructive review. We address the questions below.
>
> ## **Q1. Is Section 4 mainly a theoretical extension, or also practically useful?**
>
> We view Section 4 primarily as a theoretical extension. Its main contribution is to bring structured prediction problems previously outside the surrogate loss framework into scope by leveraging the convolutional Fenchel–Young loss. Ranking with the NDCG loss is a notable example, and other problems where the prediction space differs from the output space (e.g., ranking with prediction-at-$k$ loss in Blondel 2019, Appendix A) fall into this category.
>
> As for the practical side, the advantage of the convolutional Fenchel–Young loss itself is yet to be fully understood; still, it is promising thanks to its desirable properties: smoothness and the linear excess-risk bound (Cao et al. 2025). Exploring the practical benefits of the convolutional Fenchel–Young loss in non-stationary online structured prediction is an important direction for future work, and we believe that our theoretical foundation serves as a basis for practical developments.
>
> ## **Q2. Why were stronger adaptive baselines not included in the experiments?**
>
> The two baselines we compare our Polyak-style OGD against—constant-step-size OGD and AdaGrad—are exactly those adopted in prior work on online structured prediction with target-loss guarantees: constant-step-size OGD follows Van der Hoeven (2020) and Sakaue et al. (2024), and AdaGrad follows Van der Hoeven et al. (2021). The experiment is designed as an ablation study: by varying only the learning rate schedule while keeping the underlying OGD framework fixed, we can clearly isolate the effect of the Polyak-style learning rate.
>
> We did not include methods designed for non-stationary online convex optimization (OCO) more broadly, such as Sword++ (Zhao et al., 2024), which is arguably the state-of-the-art for non-stationary OCO. As a side note, however, applying Sword++ to our setting does not yield a theoretical improvement for the target-loss bound (see our response to Reviewer CrL4 for details), and its implementation is somewhat heavy as it requires running $O(\log T)$ parallel experts. We agree that exploring the empirical performance of such advanced methods is an interesting direction, and we will include a discussion of this as future work in the revised version.
>
> ## **Q3. Is the Polyak-style learning rate computationally practical for large output spaces?**
>
> Yes. As discussed in Section 3.4, the key quantity to compute is $\mathbb{E}_{\hat{y}_t\sim\pi(\theta_t)}[\ell(\hat{y}_t,y_t)]$ in eq. (5). The distribution $\pi(\theta_t)$ is obtained via a Frank–Wolfe-type algorithm (e.g., Garber and Wolf 2021) over $\mathrm{conv}(\\{\rho(\hat{y}):\hat{y}\in\hat{\mathcal{Y}}\\})\subseteq\mathbb{R}^d$, each iteration of which requires only a linear optimization over the convex hull, which is tractable even when the prediction-space size $|\hat{\mathcal{Y}}|$ is exponential in $d$. Typically, an $\varepsilon$-approximate $\pi(\theta_t)$ can be computed in $O(d^2\log(d/\varepsilon))$ iterations (cf. Sakaue et al. 2024, Section 3.1). In addition, using active-set-control techniques (Beck and Shtern, 2017; Besançon et al., 2025), the support size of $\pi(\theta_t)$ is at most $O(d)$ (as implied by Carathéodory's theorem), so the expectation can be computed with $O(d)$ evaluations of $\ell(\cdot,y_t)$. Thus, the per-round cost is typically dominated by $\tilde{O}(d^2)$ linear optimization calls. Therefore, even though $|\hat{\mathcal{Y}}|$ can be exponential in $d$, the computation of the Polyak-style learning rate remains computationally practical.

---

> > ### Author Rebuttal · Reviewer_zVFu · 2026-04-04
> >
> > I thank the authors for their detailed responses. The clarifications address my concerns, and I consider them resolved. Given my low confidence on this topic, I defer to the authors’ explanations.

---

### Official Review · Reviewer_CrL4 · 2026-03-13

**Soundness:** 3
**Presentation:** 3
**Significance:** 2
**Originality:** 2
**Overall Recommendation:** 4
**Confidence:** 3

**Summary:**

This paper studies dynamic regret in structured prediction problems. The approach revisits the exploiting-the-surrogate gap technique of Van der Hoeven (2020) and obtains $O(F_T+P_T)$ guarantees, where $F_T$ is the cumulative surrogate loss of the comparator sequence and $P_T$ the path-length of the comparator sequence. The analysis leads to a novel polyak-like step-size schedule which guarantees tighter regret guarantees than the conservative worst-case fixed step-size.

**Compliance With Llm Reviewing Policy:**

Affirmed.

**Final Justification:**

The rebuttal addressed my main concerns about the feasibility of obtaining the better $\sqrt{(1+F_T)(1+P_T)}$ bound that can be obtained with self-bounding losses in OCO, and provided a reasonably convincing lower bound showing that the $F_T+P_T$ is unavoidable. I leave my rating as weak accept because it is fairly concerning that well-known standard dynamic regret analysis was unknown and considered to be non-trivial, making me less sure about the novelty framing throughout the paper.

**Key Questions For Authors:**

- Is there any reason for not applying a meta-algorithm (such as SWORD+) to tune the trade-off between
$P_T$ and $\sum_t \\|G_t\\|_F^2$ to get a $\sqrt{P_T F_T}$ bound?

- Once one decides to use self-bounding surrogates, can we not just immediately apply the results of
  Zhao et al. 2024 to get $\sqrt{(1+P_T)F_T}$ bounds?

**Limitations:**

yes

**Strengths And Weaknesses:**

## Originality/Significance

**Strengths**: The paper investigates an important extension to the usual structured prediction setting to handle non-stationarity. The fact that there is a natural Polyak-like step-size for this setting is interesting

**Weaknesses**: The primary weaknesses affecting my score are the following:
- **Upper & lower bounds.** The $F_T+P_T$ upper bound bound is very weak even compared to the usual $\sqrt{(1+P_T) F_T}$ that should be obtainable here using the standard mixture-of-experts arguments. This bound is easy to make trivial: any time **one** of $P_T$ and $F_T$ is $O(T)$, the guarantee becomes $O(T)$ whereas the $\sqrt{(1+P_T)F_T}$ bound will be $O(\sqrt{T})$ as long as one of these quantities are small, which is precisely why that bound is the focus of the literature. Moreover, the bound seems weaker than even the naive $P_T\sqrt{F_T}$ bound that would be obtained by ignoring non-stationarity altogether: this naive bound still implies $\sqrt{P_T F_T}$ if it is obtained over each sub-interval (see, e.g., Cutkosky 2020), while the $P_T+F_T$ does not seem to imply any non-trivial rate even when obtained on all intervals in a strongly-adaptive sense. So it is unclear to me why the bounds obtained are significant.


   - The bound is argued to be significant primarly via the lower bound argument, but it's not clear to me that the lower bound truly precludes the stronger $O(\sqrt{(1+P_T)F_T})$ type guarantees that are obtained in, e.g., Zhao et al. (2024), which are strictly better than the $O(P_T + F_T)$ bounds obtained here. The lower bound seems to revolve around instances in which $P_T$ and $F_T$ are on the same order, in which case $\sqrt{(1+P_T)F_T}=O(P_T + F_T)$. But in general the former bound can be much better and is significantly harder to guarantee, typically requiring a mixture-of-experts meta-learning argument on top. In fact, I do not see any reason why the usual mixture-of-experts approach wouldn't work here to get an $\sqrt{P_T F_T}$ bound.

    - It seems to me that any time the losses are self-bounding, even in general OCO, the usual gradient descent analysis with a fixed step-size would give a $P_T+F_T$ bound would it not? you have essentially $R_T(u_1,\ldots,u_T)\le \frac{1+P_T}{2\eta} + \frac{\eta}{2}\sum_t\\|\nabla\ell_t(w_t)\\|^2$, and the latter term is bounded by $\frac{\beta}{2}\sum_t\ell_t(w_t)-\ell_t^*$ for self-bounding losses, so as long as $\eta\le 1/\beta$ you can get an $P_T+F_T$ bound. So this bound seems like not a particularly significant one compared to the $O(\sqrt{(1+P_T)F_T})$ bound. Such bounds do not seem to be novel in this context, since the argument is simply applying the usual self-bounding argument after obtaining second-order gradient adaptivity.

- **Independent Interest of the Dynamic Regret Decomposition.** I feel that it is misleading to present Proposition 2.4 as if it is a novel contribution of independent interest. This result follows immediately from more general existing analyses of dynamic mirror descent which I have seen several times already in the literature (e.g., Zhao et al. (2024) or Jacobsen \& Cutkosky (2021) off the top of my head). For instance, the result is immediate from Theorem 1 in Zhao et al. (2024).

**Minor Weaknesses**: The results leverage many existing results and standard arguments (e.g., dynamic regret of mirror descent, existing results regarding Fenchel-Young losses for statistical learning), or are somewhat straight-forward extensions of existing results (e.g., Appendix F.2. shows that the key result in Lemma 4.3 is a fairly straight-forward extension of the one from Cao et al. 2025).

**References**
  - Zhao, P., Zhang, Y., Zhang, L., & Zhou, Z. (2024). Adaptivity and non-stationarity: problem-dependent dynamic regret for online convex optimization. Journal of Machine Learning Research, 25(98), 1–52.

  - Jacobsen, A., & Cutkosky, A. (2022). Parameter-free mirror descent. In P. Loh, & M. Raginsky, Proceedings of Thirty Fifth Conference on Learning Theory (pp. 4160–4211). : PMLR.

  - Cutkosky, A. (2020). Parameter-free, dynamic, and strongly-adaptive online learning. In H. D. III, & A. Singh, Proceedings of the 37th International Conference on Machine Learning (pp. 2250–2259). Virtual: PMLR.

## Presentation

The paper is easy to read throughout and clearly explained. I found it an enjoyable read.

## Soundness

I did not work through the appendix in detail but am confident about the correctness of the results. The results should be unsurprising to readers familiar with dynamic regret analysis and self-bounding losses in online learning.

---

> ### Author Rebuttal · Authors · 2026-03-25
>
> We sincerely thank the reviewer for the detailed review. We address the main concerns below.
>
> ## **Q1. On applying Sword++**
> The reviewer suggests that $O(\sqrt{(1+P_T)F_T})$ should be achievable via Sword++ (Zhao et al., 2024), making our $F_T+O(P_T)$ bound appear weak. We indeed examined this direction. However, this comparison mixes surrogate-level regret analysis with our target-loss guarantee, so the conclusion is not correct. Two points need clarification.
>
> **The $F_T$ term is inherent in target loss analyses.**
> The reviewer's argument operates at the surrogate-loss OCO-regret level. However, to bound the *target loss* $\sum_t\ell_t$, the standard approach in the line of work (Van der Hoeven, 2020; Van der Hoeven et al., 2021; Sakaue et al., 2024; Shibukawa et al., 2025), including ours, exploits the surrogate gap (Assumption 3.2), yielding:
> $$\sum_t\ell_t\le(1-\alpha)F_T+(1-\alpha)\underbrace{\Bigl(\sum_tL_t(W_t)-\sum_tL_t(U_t)\Bigr)}_{\rm OCO\ dynamic\ regret}.$$
> Thus, the $(1-\alpha)F_T$ term enters before any dynamic-regret bound is plugged in. It is therefore not an artifact specific to our analysis, and improving the OCO dynamic regret alone cannot avoid the linear $F_T$ dependence.
>
> **Sword++ does not remove the linear $P_T$ dependence.**
> Applying Sword++ to the OCO-dynamic-regret term gives $O(\sqrt{(1+P_T+F_T)(1+P_T)})$ (Zhao et al., Theorem 6). Combined with the inherent $(1-\alpha)F_T$, the target-loss bound becomes $O(F_T+P_T)$. Thus, Sword++ does not improve the target-loss bound over ours. Moreover, Sword++ incurs an extra $O(\log\log T)$ factor hidden in the bound (see proof of Theorem 4 in Zhao et al.; $\ln N=O(\log\log T)$ is treated as a constant) and requires $O(\log T)$ parallel experts.
>
> **Important note on lower bounds.** The suggested $O(\sqrt{(1+P_T)F_T})$ regret bound is explicitly ruled out by Zhao et al. (2024, Theorem 8). The correct Sword++ regret bound is $O(\sqrt{(1+P_T+F_T)(1+P_T)})$, which cannot remove the linear $P_T$ dependence. Furthermore, our $\Omega(F_T+P_T)$ target-loss lower bound (Theorem 5.1) remains valid even when $F_T\simeq T$ and $P_T=0$ (Appendix G), precluding Sword++-type bounds on the target loss.
>
> Given the above, we believe achieving the $F_T+O(P_T)$ target-loss bound with a simple OGD-based approach is a significant result.
>
> ## **Q2. On the analysis with self-bounding surrogates**
> As noted above, Sword++ does not lead to the $O(\sqrt{(1+P_T)F_T})$ bound. Below, we address the point on the self-bounding case raised under Weaknesses.
>
> The reviewer suggests that self-bounding losses with fixed step-size OGD may already yield $O(F_T+P_T)$. This argument, however, stops at the *surrogate-loss* level:
> $$(1-\eta\beta/4)\sum_tL_t(W_t)\le\frac{1+P_T}{2\eta}+(1-\eta\beta/4)F_T=O(P_T+F_T).$$
> Transferring such a bound from surrogate to *target* loss requires a delicate surrogate-gap analysis, beyond standard OCO analysis. In the stationary setting, Sakaue et al. (2024) show that various structured prediction tasks admit surrogate-gap parameters $\alpha\in(0,1)$ such that setting $\eta=4\alpha/\beta$ enables the transfer from surrogate to target loss. Our work extends this idea to the non-stationary setting and broadens its scope via convolutional Fenchel–Young losses.
>
> Furthermore, $\eta=4\alpha/\beta$ is theoretically valid but yields an overly conservative learning rate in practice. Our Polyak-style learning rate (Section 3.4) automatically adapts $\eta_t$ to be more aggressive while preserving the theoretical guarantee, constituting a practically meaningful contribution, as demonstrated in Appendix E.
>
> ## **On Proposition 2.4**
> We do not think Proposition 2.4 follows immediately from Theorem 1 of Zhao et al. (2024). While our starting decomposition is obtainable from Theorem 1 by setting $M_t=0$ with the squared-L2 regularizer, this is only the first step. The nontrivial part is bounding the non-stationarity term
> $$\sum_t\frac{\\|W_t-U_t\\|^2-\\|W_{t+1}-U_t\\|^2}{2\eta_t}$$
> by $O((1+P_T)/\eta_T)$ when $\eta_t$ is *non-increasing*. This contrasts with the existing analysis; for example, Zhao et al. (2024, Lemma 1) handle the term only for a *constant* step size, where the bound reduces to a straightforward telescoping. Our proof in Appendix D extends this to non-increasing $\eta_t$ via more delicate summation by parts and telescoping. If the reviewer is aware of an existing reference containing this analysis, we would greatly appreciate the pointer.
>
> That said, Proposition 2.4 is a supporting tool rather than a main novelty claim. Its role is to enable the Polyak-style learning rate to be used with theoretical guarantees, as in Section 3.4.
>
> ## **On minor weaknesses**
> Regarding the connection to Cao et al. (2025), Lemma 4.3 is related to their Lemma 14, but they are distinct as explained in Appendix F.2. More importantly, our Lemma 4.4 reveals the self-bounding-like property of the convolutional Fenchel–Young loss, which is not covered by Cao et al. (2025).

---

> > ### Author Rebuttal · Reviewer_CrL4 · 2026-04-02
> >
> > **upper bounds:** Interesting, thank you for the clarification; you are correct that
> > SWORD++ would actually imply $F_T+P_T$ in this context.
> >
> > **lower bounds:** $\sqrt{(1+P_T)F_T}$ ruled out by Zhao (2024) theorem 8
> >
> > The lower bound of Theorem 8 shows that the regret is $O(\sqrt{(1+F_T)(1+P_T)})$. This does not rule
> >    out what I was suggesting above, and my example still applies. This nuance, that they rule
> >    out $\sqrt{F_T(1+P_T)}$, is merely saying that you can't have 0 regret by having $F_T$ be zero, which
> >    is not the issue my comment was pointing out. My example holds even with the $+1$ offset to $F_T$: you can still have
> >    $F_T+P_T$ being $O(T)$ while $F_T+\sqrt{(1+F_T)(1+P_T)}$ is $O(\sqrt{T})$. So this discrepency is not ruled out by
> >    the lower bound of Zhao (2024).
> >
> > **Self-bounding surrogates:** I'm not sure I see the point being made. In the answer to Q1 it is noted that
> > the target losses are bounded by $(1-\alpha)F_T + (1-\alpha)\sum_t L_t(W_t)-L_t(U_t)$, so you can apply the usual self-bounding
> > arguments for the latter term, yielding $(1-\alpha)F_T + O(F_T+P_T)=O(F_T+P_T)$, which again would suggest that
> > this type of bound is straight-forward any time you have the self-bounding property
> >
> > **Novelty of the adaptive step-size:** I maintain that this is not novel. It is indeed standard, particularly for
> > the non-increasing step-size case, which is the standard case for time-varying step-sizes.
> > The following steps are entirely standard:
> >
> > \begin{align*}
> > \sum_t \frac{\\|W_t-U_t\\|^2}{2\eta_t} - \frac{\\|W_{t+1}-U_t\\|^2}{2\eta_t}
> > &=
> > \sum_t \frac{\\|W_t-U_t\\|^2}{2\eta_t} - \frac{\\|W_{t+1}-U_t\\|^2}{2\eta_{t+1}} + \sum_t \left(\frac{1}{2\eta_{t+1}}-\frac{1}{2\eta_t}\right)\\|U_t-W_{t+1}\\|^2\newline
> > &\le
> > \sum_t \frac{\\|W_t-U_t\\|^2}{2\eta_t} - \frac{\\|W_{t+1}-U_t\\|^2}{2\eta_{t+1}}+ \frac{D^2}{2\eta_{T+1}}
> > \end{align*}
> > The first summation is
> > \begin{align*}
> > S
> > &=
> > \frac{\\|W_1-U_1\\|^2}{2\eta_1}-\frac{\\|W_{T+1}-U_T\\|^2}{2\eta_{T+1}}+\sum_{t=2}^T \frac{\\|W_t-U_t\\|^2}{2\eta_t} - \frac{\\|W_{t}-U_{t-1}\\|^2}{2\eta_{t}}\newline
> > &\le
> > \frac{\\|W_1-U_1\\|^2}{2\eta_1}-\frac{\\|W_{T+1}-U_T\\|^2}{2\eta_{T+1}}+\sum_{t=2}^T \frac{\\| W_t-U_t\\|\\|U_t-U_{t-1}\ \|}{\eta_t}\newline
> > &\le
> > \frac{\\|W_1-U_1\\|^2}{2\eta_1}-\frac{\\|W_{T+1}-U_T\\|^2}{2\eta_{T+1}}+\frac{DP_T}{\eta_{T+1}}.
> > \end{align*}
> > which is again a standard computation. These are the steps referenced directly in Zhao 2024 remark 1(ii). Such computations can be seen
> > very frequently throughout the literature. One reference which shows it more explicitly
> > than Zhao et al. (2024) would be Hall \& Willet (2015)
> > Theorem 2,
> > which admits this bound as a special case of a more general path-length bound.
> >
> > - E. C. Hall and R. M. Willett, "Online Convex Optimization in Dynamic Environments," in IEEE Journal of Selected Topics in Signal Processing, vol. 9, no. 4, pp. 647-662, 2015.
> >
> >
> > Overall I am not convinced that $F_T+P_T$ is indeed optimal, and seems to be achievable using standard self-bounding arguments. Moreover, the insistence on novelty of the dynamic regret decomposition weakens my confidence in the paper's command of prior work and the reliably of the novelty framing throughout.

---

> > > ### Author Response · Authors · 2026-04-03
> > >
> > > We sincerely thank the reviewer for the careful follow-up and for clarifying that the original upper-bound concern was resolved. We also appreciate the pointer regarding Proposition 2.4; we agree that our initial response overlooked the existing literature, and we will revise the wording and citations accordingly.
> > >
> > > To avoid notation confusion, let us first clarify: $\ell_t=\mathbb E[\ell(\hat y_t,y_t)]$ denotes the learner’s expected target loss at round $t$ (the expected 0-1 loss in the case of Appendix G), while $L_t(W)$ is the surrogate loss at $W$.
> > >
> > > ## **Lower bounds**
> > > Theorem 8 of Zhao et al. (2024) shows $\sum_t(L_t(W_t)-L_t(U_t))=\Omega(\sqrt{(1+F_T)(1+P_T)})$ for the regret of convex smooth functions $L_t$, not for our target loss $\sum_t\ell_t$ (neither convex nor smooth). Our lower bound (Theorem 5.1, Appendix G) already shows $\sum_t\ell_t=\Omega(T)$ even when $(F_T,P_T)=(O(T),0)$ or $(0,O(T))$, which means the reviewer's initial guess "the lower bound seems to revolve around instances in which $F_T$ and $P_T$ are on the same order" does not apply here. This also shows that the suggested bound of $O(F_T+\sqrt{(1+F_T)(1+P_T)})$ does not apply to our target loss $\sum_t\ell_t$, since it reduces to $\sum_t\ell_t=O(\sqrt{T})$ when $(F_T,P_T)=(0,O(T))$, contradicting $\sum_t\ell_t=\Omega(T)$.
> > >
> > > Moreover, slightly refining the proof of Theorem 5.1 yields
> > > $$\sum_t\ell_t=\Omega(F_T+P_T)$$
> > > for any values of $F_T,P_T\in[0,T/2]$ up to constant factors. Thus, our $\sum_t\ell_t=O(F_T+P_T)$ bound is asymptotically tight in this target-loss sense. The proof can be refined as follows.
> > >
> > > ---
> > > **Refined proof of Theorem 5.1.**
> > > Consider a simple block construction on top of the Appendix G instance. Let $f,p>0$ be any integers with $f+p\le T$ and use the random binary classification instance given in Appendix G over the first $f+p$ rounds, so the learner suffers $\sum_t\ell_t\ge(f+p)/2$. Define the comparator sequence as follows.
> > >
> > > For the first $f$ rounds, use the constant comparator
> > > $$U_t^{\rm F}=\begin{bmatrix}0&1/\sqrt{2}\\\\0&1/\sqrt{2}\end{bmatrix}.$$
> > > Since Appendix G uses $x_t=(1,0)^\top$, we have $U_t^{\rm F}x_t=(0,0)^\top$ and hence $\zeta_t=0$, so the smooth hinge loss $L_t(U_t)$ is exactly $1$ in these rounds.
> > >
> > > For the next $p$ rounds, use
> > > $$U_t^{\rm P}=\begin{bmatrix}y_t/2&(-1)^t/\sqrt{2}\\\\-y_t/2&0\end{bmatrix},$$
> > > where $y_t\in\\{-1,+1\\}$ is the ground-truth label at round $t$. Because the first column is exactly that of case (i) in Appendix G, we have $\zeta_t=1$, so $L_t(U_t)$ is exactly $0$ in these rounds.
> > >
> > > For all $t>f+p$, repeat the last round of the second block, i.e., set $y_t=y_{f+p}$ and $U_t=U_{f+p}^{\rm P}$.
> > >
> > > Consequently, this comparator sequence gives $F_T=\sum_tL_t(U_t)=f$. As for the path length, the boundary step between the two blocks and the $p-1$ transitions inside the second block give $\sqrt2(p-1)+1≤P_T≤2p$ by direct calculation, hence $P_T=\Theta(p)$. Since we have $\sum_t\ell_t\ge(f+p)/2$, $F_T=f$, and $P_T=\Theta(p)$ for any integers $f,p>0$ with $f+p≤T$, we obtain $\sum_{t=1}^T\ell_t=\Omega(F_T+P_T)$ for any $F_T,P_T\in[0,T/2]$ up to constant factors.
> > >
> > > ---
> > > Thank you for the helpful follow-up, which made us realize that this refined lower bound should be stated explicitly in the revised version. This clarifies that our $O(F_T+P_T)$ bound is tight up to constants, and we hope this addresses the reviewer's main remaining concern.
> > >
> > > ## **Self-bounding surrogates**
> > > Our response to Q1 intentionally showed the simplified inequality to focus on how Sword++ could be applied; in doing so, we omitted the surrogate-to-target transfer, which may have made the $O(F_T+P_T)$ bound appear straightforward. That transfer is exactly the missing step: a surrogate-level self-bounding argument alone does not give a target-loss guarantee. Our contribution is to show that the transfer originally developed for the stationary setting—exploiting the surrogate gap (Van der Hoeven, 2020; Sakaue et al., 2024)—works against non-stationary comparator sequences, which is not obvious a priori.
> > >
> > > Concretely, the key issue is not merely analyzing the OCO dynamic regret, but identifying a target-side cancellation that removes the learner’s cumulative surrogate loss $\sum_tL_t(W_t)$ from the final target-loss bound. In our notation, the $\eta_t\\|G_t\\|^2/2$ term must be absorbed by the per-round *surrogate–target gap* $L_t(W_t)-\ell_t$, which is exactly what Eq. (4)/(8) encodes. This does not follow from the standard OCO regret analysis for self-bounding surrogate losses $L_t$. Once this target-side derivation is identified, using a small constant learning rate $\eta$ is justified, but this is too conservative in practice. The additional point is that the transfer remains valid under our more aggressive Polyak-style learning rate $\eta_t$, which empirically performs well, particularly in non-stationary environments (Appendix E). These are our main technical contributions for non-stationary online structured prediction.

---

### Official Review · Reviewer_2pmD · 2026-03-23

**Soundness:** 4
**Presentation:** 4
**Significance:** 4
**Originality:** 3
**Overall Recommendation:** 5
**Confidence:** 1

**Summary:**

this paper studies full-information online structured prediction. previous online structure prediction studies focus mainly on surrogate regret and try to get a finite bound independent of T. however, in non-stationary environments, every fixed estimator may incur surrogate loss that grows linearly with T, making the existing theory invalid. to overcome this limitation, the contributions of this paper include: (1) proposed a new upper bound (2) combined the dynamic regret analysis of online gradient descent + exploit-the-surrogate-gap (3) extend their approach to broader settings beyond prior work via the convolutional Fenchel–Young loss (4) proved that the dependence on F_T and P_T is tight.

**Compliance With Llm Reviewing Policy:**

Affirmed.

**Final Justification:**

I would maintain my score (accept) if the authors would lower the tone about complex structures in the introduction.

**Key Questions For Authors:**

can you extend the experiments to more extensive real datasets?

what's the computational cost of computing the Polyak style learning rate when handing a large prediction space? does it scale to high dimensions?

in practice, how to determine lambda?

**Limitations:**

yes. the authors discussed the limitation of the full information setup and mentioned the bandit feedback.

**Strengths And Weaknesses:**

the authors successfully combined the dynamic regret analysis of OGD with the exploit-the-surrogate-gap technique. the proofs of theorems 3.4 and 4.5 look clear and logically sound. the assumptions (self bounding property) and the surrogate gap condition seem to align with what is commonly assumed in this area.

weaknesses: the datasets in the experiment section are synthetic and MNIST. although the result shows that the superiority of Polyak style learning rate, however, it doesn't seem to show the performance on a complex structured object, such as a matching or a tree, as mentioned in the intro.

---

> ### Author Rebuttal · Authors · 2026-03-25
>
> We sincerely thank the reviewer for the positive and constructive review. We address the questions below.
>
> ## **Q1. Experiments on complex structured objects and real datasets**
> We appreciate this suggestion. Our current experiments focus on synthetic binary classification and MNIST multiclass classification. While we have not yet conducted experiments on complex structured objects (e.g., matching or trees), we note that the advantage of the Polyak-style learning rate is consistently observed across all settings: the more non-stationary the environment, the larger the benefit. This trend is expected to carry over to other structured prediction problems, because the learning rate primarily affects online convex optimization of surrogate losses, while the prediction space $\hat{\mathcal{Y}}$ primarily matters in the decoding step; the presence of complex structures mainly affects the latter. For now, we leave the extension of experiments to a wider range of real-world structured prediction tasks for future work. We will add a discussion on this point in the revised version.
>
> ## **Q2. Computational cost of the Polyak-style learning rate for large prediction spaces**
> The Polyak-style learning rate in eq. (5) requires computing $\mathbb{E}_{\hat{y}_t\sim\pi(\theta_t)}[\ell(\hat{y}_t,y_t)]$, where $\pi(\theta_t)$ is the randomized decoding distribution. This constitutes the dominant computational factor and, as discussed in Section 3.4, can be done efficiently even when $|\hat{\mathcal{Y}}|$ is exponential in $d$. Specifically, $\pi(\theta_t)$ is obtained via a Frank–Wolfe-type algorithm (e.g., Garber and Wolf 2021) run over $\mathrm{conv}(\\{\rho(\hat{y}):\hat{y}\in\hat{\mathcal{Y}}\\})\subseteq\mathbb{R}^d$, each iteration of which requires only a linear optimization over $\mathrm{conv}(\\{\rho(\hat{y}):\hat{y}\in\hat{\mathcal{Y}}\\})$. Typically, the number of iterations required to compute $\varepsilon$-approximate $\pi(\theta_t)$ is $O(d^2\log(d/\varepsilon))$ (cf. Sakaue et al. 2024, Section 3.1). Combined with active-set-control techniques (Beck and Shtern, 2017; Besançon et al., 2025), the support size of $\pi(\theta_t)$ is at most $O(d)$ (as guaranteed by Carathéodory's theorem), so the expectation in eq. (5) can be computed by evaluating $\ell(\cdot,y_t)$ $O(d)$ times. Therefore, ignoring logarithmic factors, the per-round cost is typically dominated by the $\tilde O(d^2)$ calls to the linear optimization oracle over $\mathrm{conv}(\\{\rho(\hat{y}):\hat{y}\in\hat{\mathcal{Y}}\\})$. This is much more scalable than the prediction-space size $|\hat{\mathcal{Y}}|$, which is generally exponential in $d$.
>
> In some typical problems, the computation cost can be further reduced. For $d$-class classification with the logistic loss, for example, $\pi(\theta_t)$ can be obtained by computing the softmax of $\theta_t$, which can be done in $O(d)$ time.
>
> ## **Q3. How to determine $\lambda$ in practice**
> The parameter $\lambda$ represents the strong convexity modulus of $\Omega$ (Assumption 4.2) and is determined by the choice of $\Omega$. In practice, natural choices of $\Omega$ come with a known constant $\lambda$. For example, taking $\Omega$ to be the negative Shannon entropy (as in the logistic loss) yields $\lambda=1$ with respect to the $\ell_1$-norm. More generally, one can always ensure $\lambda$-strong convexity by adding a $\lambda$-strongly convex regularizer to the indicator of $\mathcal{C}$ (as noted in Section 4). A similar principle applies to the $\lambda$ in Section 3, where it enters through the surrogate gap parameter $\alpha=4\gamma/(\lambda\nu)$.
>
> To clarify, the cautionary note in Section 4.3 is meant to address tighter bounds via adaptive control of $\lambda$, while our Theorem 4.5 already gives target-loss guarantees for any fixed $\lambda$ satisfying Assumption 4.2.

---

> > ### Author Rebuttal · Reviewer_2pmD · 2026-04-02
> >
> > The authors addressed my questions but would be great if the authors lower the tone about complex structures in the introduction.

---

### Decision · Program_Chairs · 2026-04-30

**Decision:**

Accept (regular)

**Comment:**

This paper receives uniformly positive reviews. It is not easy to identify such a natural problem in the already crowded area of non-stationary online prediction with full information, particularly within the path-length literature. The binary-classification motivation illustrates how additional structure should be leveraged in this broad line of work. The introduction of the cumulative surrogate loss $F_T$ alongside the path length $P_T$ is fresh, and the matching upper and lower bounds are tight in all these parameters. I recommend accept.